# The Sun and Space Weather

Nat Gopalswamy 

NASA Goddard Space Flight Center, Greenbelt, MD 20771, USA; nat.gopalswamy@nasa.gov

**Abstract:** The explosion of space weather research since the early 1990s has been partly fueled by the unprecedented, uniform, and extended observations of solar disturbances from space- and ground-based instruments. Coronal mass ejections (CMEs) from closed magnetic field regions and high-speed streams (HSS) from open-field regions on the Sun account for most of the disturbances relevant to space weather. The main consequences of CMEs and HSS are their ability to cause geomagnetic storms and accelerate particles. Particles accelerated by CME-driven shocks can pose danger to humans and their technological structures in space. Geomagnetic storms produced by CMEs and HSS-related stream interaction regions also result in particle energization inside the magnetosphere that can have severe impact on satellites operating in the magnetosphere. Solar flares are another aspect of solar magnetic energy release, mostly characterized by the sudden enhancement in electromagnetic emission at various wavelengths—from radio waves to gamma-rays. Flares are responsible for the sudden ionospheric disturbances and prompt perturbation of Earth's magnetic field known as magnetic crochet. Nonthermal electrons accelerated during flares can emit intense microwave radiation that can drown spacecraft and radar signals. This review article summarizes major milestones in understanding the connection between solar variability and space weather.

**Keywords:** solar eruptions; solar flares; coronal mass ejections; geomagnetic storms; solar energetic particle events; coronal holes; corotating interaction regions

## 1. Introduction

The Sun is an ordinary star from an astronomical point of view, but it is the vital source of energy that supports life on Earth. Due to its proximity to Earth, we can observe and understand Sun's variability on various timescales, from sub-second to centuries. Most of the variability is caused by solar magnetism, thought to operate in the outer shell of the Sun. Observationally, the variability manifests as the appearance and dispersal of bipolar magnetic regions (e.g., sunspot regions) and unipolar regions (coronal hole regions). The solar dynamo is sustained by the exchange between toroidal flux, represented by sunspots, and the poloidal flux, represented by polar field strength. Solar eruptions are part of the life cycle of active regions (ARs), in that photospheric motions store energy in AR magnetic fields, and the stored energy is explosively released. Coronal mass ejections (CMEs) and solar flares are two manifestations of the energy release from closed magnetic regions. Coronal holes contain field lines open to the space, and the solar plasma can readily escape into space as high-speed streams (HSS). Thus, the two types of magnetic topology on the Sun result in two types of mass emission: CMEs and HSS. As CMEs and HSS propagate into the corona and interplanetary (IP) space, they interact with the ambient solar wind forming shock sheaths ahead of CMEs and stream interaction regions (SIRs) at the leading edge of HSS. A solar flare represents a transient increase in electromagnetic emission at all wavelengths from radio to gamma rays originating from localized closed magnetic field regions on the Sun. The flare emissions are caused by nonthermal electrons (radio bursts and hard X-ray bursts) and protons (impulsive gamma rays) energized in the magnetic reconnection region in the active region corona. The accelerated particles precipitating to the chromosphere cause chromospheric evaporation, and the heated flare plasma emits in

soft X-rays. Magnetic reconnection results simultaneously in a post-eruption arcade (PEA) and a magnetic flux rope (FR). Thermal emission from the PEA in soft X-rays is used as an indicator of flare strength. The FR is accelerated outwards as long as the reconnection proceeds, followed by interaction with the ambient solar wind. If a FR is fast enough, it drives a fast-mode magnetohydrodynamic (MHD) shock that can accelerate particles to GeV energies. Such particles are known as solar energetic particles (SEPs). Flares may also contribute to SEPs. SIRs also accelerate particles to lower energies, typically beyond 1 au. All these phenomena—flare electromagnetic emission, CMEs, SEPs, CIRs, and HSS— can contribute to adverse space weather in the heliosphere. Space weather effects can be felt in Earth's magnetosphere, ionosphere, atmosphere, and surface when Earth is in the path of these disturbances. Therefore, forecasting the properties of these disturbances and their arrival time at Earth is important for space weather prediction. There has been significant progress in understanding how solar eruptions result in various space weather consequences over the past two decades, as reviewed recently [1–7].

The purpose of this paper is to summarize the observational results on solar disturbances and highlight some key results relevant to space weather. The paper is organized as follows: Section 2 provides an update on the basic properties of CMEs. Section 3 highlights the shock-driving capability of CMEs. Section 4 focuses on the CME origin of SEP events. Section 5 summarizes the CME link to geomagnetic storms via the southward magnetic field component and the speed of CMEs. Section 6 highlights those CME properties that seem to be essential for the acceleration of SEPs. Section 7 discusses the spacecraft anomalies that follow SEP events and geomagnetic storms. Section 8 summarizes the solar cycle (SC) variation of the CME rate and how it affects space weather events. Section 9 describes the extreme events of SC 23. Concluding remarks are given in Section 10.

## 2. Basic Properties of CMEs

### 2.1. Morphological Properties

CMEs appear as excess material newly appearing in the corona and moving away from the Sun. Although white-light coronagraphs are the most commonly used instruments to detect CMEs, extreme ultraviolet (EUV) imagers have become key instruments in observing CMEs closer to the solar surface. The combination of inner coronal images in EUV or other wavelengths, and white-light images, provide the full picture of CMEs early in their life. A CME typically starts with a slightly larger initial size than the solar source, grows in size as it expands, and becomes a large-scale coherent structure in the coronagraphic field of view. A typical CME has a number of substructures with different densities, temperatures, and magnetic field strengths [8]. CMEs often show a three-part structure consisting of a bright front followed by a dark void and a bright core [9,10]. The bright front is the compressed coronal material caused by the outward motion of the dark void interpreted as a magnetic flux rope, while the bright core in the CME interior is the eruptive prominence. The three-part structure is somewhat of an incomplete description when it comes to shock-driving CMEs: a shock forms ahead of the bright front with a compressed sheath behind the shock. Although the shock is too thin to be discerned from white-light observations, the sheath is identified as a diffuse feature surrounding the bright front [11–16].

The CME in Figure 1 has all the substructures: shock sheath, bright front, void, and core observed by the Large Angle and Spectrometric Coronagraph (LASCO) on board the Solar and Heliospheric Observatory (SOHO) mission. The bright front in Figure 1b is thought to indicate the outline of a magnetic flux rope identified with the void region. The presence of a shock can be inferred from the kink (marked S in Figure 1b) in the nearby streamer. The shock sheath can be seen better in the difference (event minus pre-event) image in Figure 1c as a diffuse structure surrounding the CME flux rope [17]. The outer edge of the sheath is taken as the shock, because the shock is too thin compared to the spatial resolution of the coronagraph images. Different substructures have different space weather consequences. When the sheath and/or the flux rope contains a south-pointing, out-of-the-ecliptic magnetic field component, a geomagnetic storm ensues upon arrival at

the magnetosphere. The bright prominence core has high density, and so it can enhance a geomagnetic storm if the density enhancement occurs during an interval in which the CME flux rope has a southward component (see, e.g., [18–20]). On the other hand, the outermost structure, viz., the CME-driven shock, is responsible for accelerating particles to high energies (e.g., [21,22]).

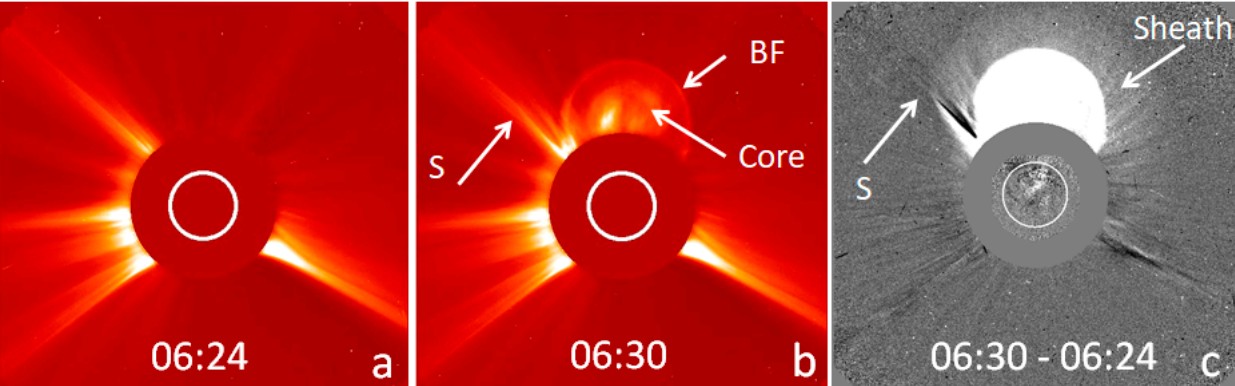

**Figure 1.** SOHO/LASCO/C2 images of the 15 January 2005 CME showing the basic morphology. The 06:24 UT image (**a**) shows the pre-eruption corona featuring several streamers. The white circle on the image of the coronagraph occulting disk denotes the optical Sun. Six minutes later, a CME appears in the north–northwest part of the corona with a bright front (BF) and bright core (**b**). The void can be seen separating BF and the core. "S" denotes a kink that appears outside of the BF. The difference between the frames (**a**,**b**) shows the coronal changes occurring over a period of 6 min (**c**). The kink S is at the outer boundary of the diffuse structure that envelopes BF. An EUV difference image, superposed on the LASCO difference image, shows disturbances on the solar disk indicating the eruption (adapted from [17]).

*2.2. Physical Properties*

CMEs are magnetized plasmas formed out of closed magnetic field regions. The CME core is the eruptive prominence and hence has the lowest temperature ($\sim 10^4$ K, [23–25]). The flux rope forms out of the sheared loops in the corona, so the temperature should be coronal ($\sim 2$ MK). The flux rope is a low-beta entity (the magnetic pressure is much larger than the thermal pressure), and the magnetic field strength is well above that in the ambient medium. The shock sheath consists of the ambient plasma compressed by the shock, so the temperature, density, and magnetic field strength are all higher than those in the ambient medium. Interplanetary CMEs (ICMEs) do confirm the basic spatial structure with an often well-defined shock, sheath, and driving flux rope. The intervals of high-density prominence material are the coolest within MCs and show low Fe and O charge states [26–36]. Recent statistics indicates that about 36% of MCs possess prominence material, indicated by the unusual O5+ and/or Fe6+ abundances [36]. In most cases, the prominence material is located at the back end of MCs, consistent with the spatial ordering observed near the Sun. However, there are reports on filament material located in the front of MCs [34,37,38]. In most of the 1-au flux ropes, heavy ions are in high-charge states, indicating hot plasma entering from the flare site into the flux rope and the charge states are frozen soon after the entrance [39]. Interestingly, both magnetic cloud (MC) and non-cloud ICMEs show charge-state enhancement, indicating that both types of ICMEs have flux rope structures paired with post-eruption arcades formed in the reconnection process [40]. Furthermore, Marubashi et al. [41] have shown that a flux rope can be fit to most of the non-cloud ICMEs with slight changes in the ICME boundaries.

*2.3. Kinematic Properties*

Figure 2 shows the speed and width distributions of CMEs detected in the SOHO/LASCO FOV (2.5 to 32 Rs) covered by the C2 and C3 telescopes. The speed of CMEs measured in

the sky plane varies by over two orders of magnitude from ~50 to > 3000 km s$^{-1}$, while the width ranges from <20° to >120°. The typical speed and width are ~400 km/s and ~40°, respectively. The speed is lognormally distributed, which has been attributed to the complexities of the elementary reconnection processes during an eruption [42]. The CME widths > 120° are mainly due to projection effects. The last width bin corresponds to full halo CMEs, which constitute only ~2.5% of all CMEs. For a given coronagraph, halo CMEs represent an energetic population with the inherent width and speed larger than the average values shown in Figure 2 [43]. The accelerations have a large scatter, but there is a clear tendency for faster CMEs to decelerate on average. On the other hand, very slow CMEs (speed < 480 km/s) have a positive acceleration. However, there are many fast CMEs that do have positive acceleration within the LASCO FOV. All CMEs have to accelerate from zero speed, so the initial acceleration is always positive. What is shown in Figure 2 is the residual acceleration after the CMEs have attained their peak speeds, and hence the initial acceleration is often missed. Observations from SOHO's Extreme-ultraviolet Imaging Telescope (EIT) and LASCO's C1 telescope reveal the extent of the initial acceleration. Case studies that include CME motion below the LASCO/C2 occulting disk reveal the CME initial acceleration, which is much higher than the residual acceleration [44–47]. Figure 3 shows the 11 June 1998 CME observed close to the Sun by SOHO/EIT and LASCO. The height–time plot of the CME has an S-shape because of the initial positive acceleration and later deceleration. If we use LASCO/C2 and C3 data alone, we see only the decelerating part (https://cdaw.gsfc.nasa.gov/CME_list/UNIVERSAL/19 98_06/htpng/19980611.102838.p097g.htp.html, accessed on 1 October 2022). The residual acceleration is ~36 m s$^{-2}$. The two EIT and two LASCO/C1 data points are able to capture the initial positive acceleration, which is an order of magnitude higher than the residual acceleration. Unfortunately, SOHO/LASCO/C1 ceased operations in June 1998. The COR1 coronagraph on board the Solar Terrestrial Relations Observatory (STEREO) [48] has observed CMEs closer to the Sun since 2006, and the acceleration has been confirmed to be in the range 0.02 to 6.8 km s$^{-2}$ using a larger number of events [49]. Studies of initial acceleration have shown that CMEs attain peak acceleration within ~1.5 Rs; the peak acceleration is inversely proportional to the duration of acceleration [47,49]. The source regions that produce high impulsive acceleration are compact compared to those that produce small gradual acceleration.

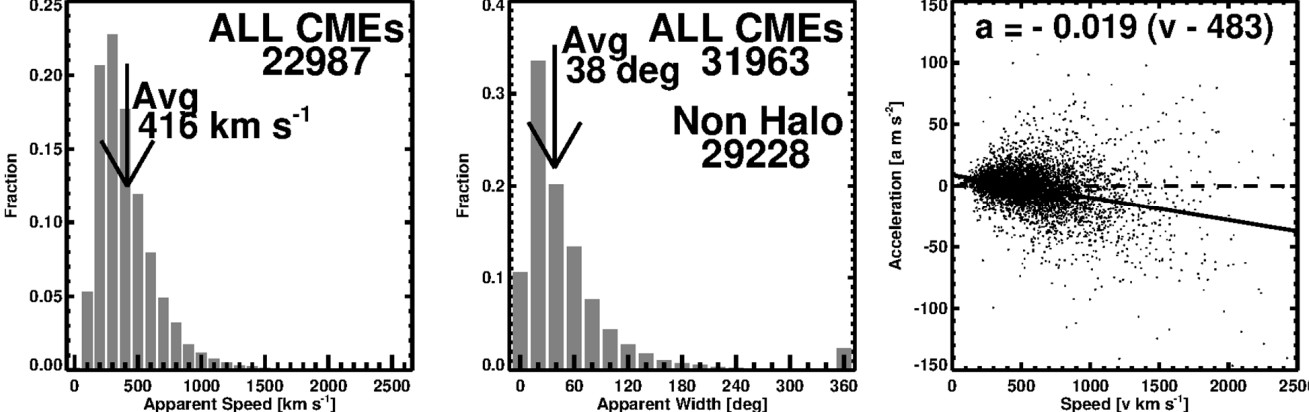

**Figure 2.** Speed (**left**) and width (**middle**) distributions of CMEs detected by SOHO/LASCO from 1996 to November 2021. (**right**) A scatter plot between CME speed and the acceleration within the LASCO FOV. The number of CMEs is different between the speed and width panels because speed measurements are not possible in many events.

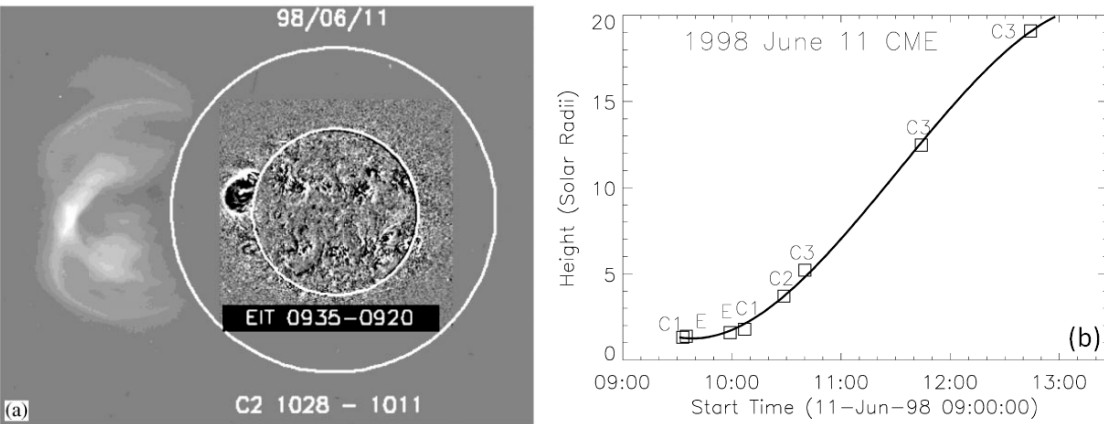

**Figure 3.** (**a**) The 11 June 1998 CME at two different instances: in the SOHO/EIT FOV at 09:35 UT and in the LASCO/C2 FIV at 10:28 UT. The CME was also observed in the LASCO/C1 FOV (not shown). The inner and outer white circles represent the solar disk and the LASCO/C2 occulting disk, respectively. (**b**) CME height–time plot that combines SOHO/LASCO (C1, C2, C3) and EIT (E) observations. The solid curve is a third-order polynomial fit to the data points (adapted from [46]).

### 2.4. CME Mass and Kinetic Energy

By estimating the number of coronal electrons from the observed brightness of CMEs, one can derive the number of protons and ions in the corona associated with these electrons and hence the mass of CMEs [50]. The left column of Figure 4 shows the CME mass distribution. The mass ranges over five orders of magnitude from ~$10^{12}$ g to ~$10^{17}$ g, with an average value of $3.5 \times 10^{14}$ g [51]. From the mass and average speed of each CME in the LASCO FOV, we can obtain the CME kinetic energy, which is shown in the right column of Figure 4. The kinetic energy varies over seven orders of magnitude from ~$10^{26}$ erg to ~$10^{33}$ erg, with an average value of $1.7 \times 10^{29}$ erg. For limb CMEs, one can determine the speed and mass without projection effects. The bottom panels of Figure 4 show the mass and kinetic energy of ~1100 limb CMEs. We see that the average mass ($1.7 \times 10^{15}$ g) and kinetic energy ($2.2 \times 10^{30}$ g) are higher by an order of magnitude compared to the general case, although the ranges remain the same. These values are consistent with those of the pre-SOHO CMEs [52,53].

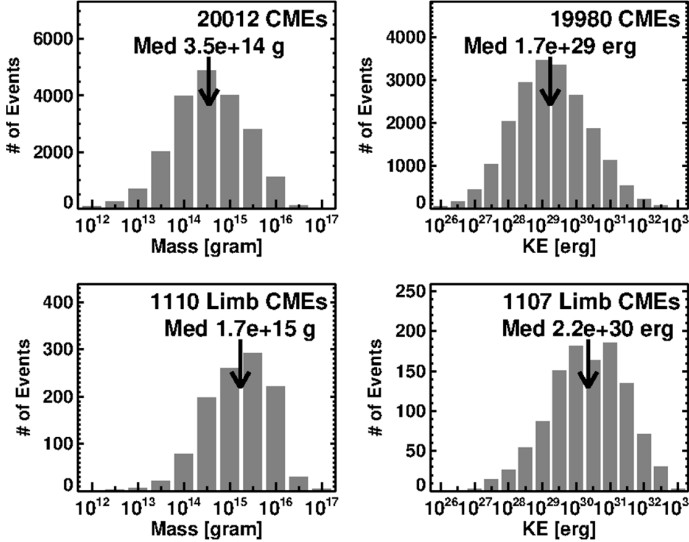

**Figure 4.** # means Number in Figure 4. Mass (**left column**) and kinetic energy (**right column**) of SOHO/LASCO CMEs from January 1996 to December 2020. The upper (**lower**) panels correspond to all (**limb**) CMEs.

It is well-known that faster CMEs are wider [54,55]. Considering limb CMEs from SC 23, Gopalswamy et al. [55] found that the CME width W and speed V are reasonably correlated: W = 0.11 V + 24.3 (W in degrees and V in km/s). Wider CMEs are also more massive [56,57]: log M = 12.6 + 1.3 log W (M in g and W in degrees). Thus, fast and wide CMEs are more energetic. This is an important characteristic, because energetic CMEs are able to travel far into the IP medium and contribute to severe space weather. The mass loss from the Sun due to CMEs amounts to ~10% of the mass loss due to the solar wind [58,59]. In a recent investigation, Michalek et al. [57] reported that wider CMEs contribute significantly to the Sun's mass loss: halo and partial halo CMEs contribute ~20% each, while CMEs with widths in the range 50–120° contribute ~60%.

## 3. CME Source Regions, Flares, and Filaments

The CME kinetic energy $>10^{33}$ erg has to be of magnetic origin [60]. Such huge amounts of energy can be stored and released in closed magnetic regions, such as sunspot regions. Closed magnetic fields also occur in non-spot regions called filament regions. Figure 5 shows examples of closed magnetic field regions at the three layers of the solar atmosphere: the photosphere, chromosphere, and corona. The compact magnetic regions are active regions (sunspot regions, but sometimes they can occur without a sunspot). One can identify five of them in Figure 5a, including the region marked A. Other regions are quiescent filament regions such as B, which also have enhanced but weaker magnetic fields than in active regions, and they are spatially more extended. Powerful CMEs can originate from both types of closed-field regions. Before eruption, bright loops can be found in the corona in both types of regions, although the active region loops are much brighter. At least some of these loops are incorporated into an erupting flux rope.

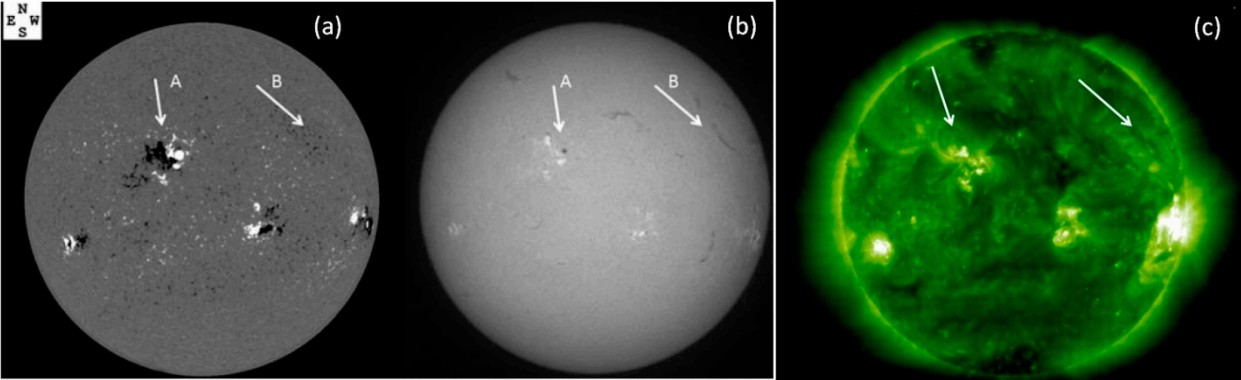

**Figure 5.** A photospheric magnetogram from the Big Bear Solar Observatory (**a**), chromospheric (H-alpha) image from the Kanzelhöhe Observatory (**b**), and a coronal (EUV) image from SOHO/EIT (**c**), all obtained on 13 May 2005. A large active region (A) and a filament region (B) are marked by arrows. In (**a**), black and white correspond to negative and positive magnetic polarity, respectively. In region A, the H-alpha image shows a sunspot, a reverse-S shaped dark filament, and bright plages, while in the region B the image shows just a dark filament. In the EUV images, active regions are bright compact features. Even the filament region shows surrounding faint loop arcade. The dark features in the EUV image are coronal holes.

Figure 6 shows an eruption from a weak field region (quiescent filament region). The region contains a horizontal filament, as imaged by the Nobeyama Radioheliograph (NoRH) at the heliographic location S54E46. In a subsequent image taken later in the day, the filament has disappeared. Such events are known as disappearing solar filament (DSF) events. In the corresponding EUV images, one can see similarities in the pre-eruption image, but after the filament disappears, there is diffuse brightening surrounding the original location of the filament. The diffuse brightening is the PEA. The erupted filament can be seen as the core of the associated CME, as shown in the 17:48 UT LASCO image. The

CME first appears in the LASCO FOV at 15:28 UT and has an average speed of 293 km/s. The CME accelerates throughout its passage of LASCO FOV, with an average acceleration of 15.9 m s$^{-2}$, attaining a speed of 524 km/s by the time it reaches ~15 Rs. The PEA is so faint that it cannot be discerned in the GOES soft X-ray light curve, even though the background is very low (~A4.5).

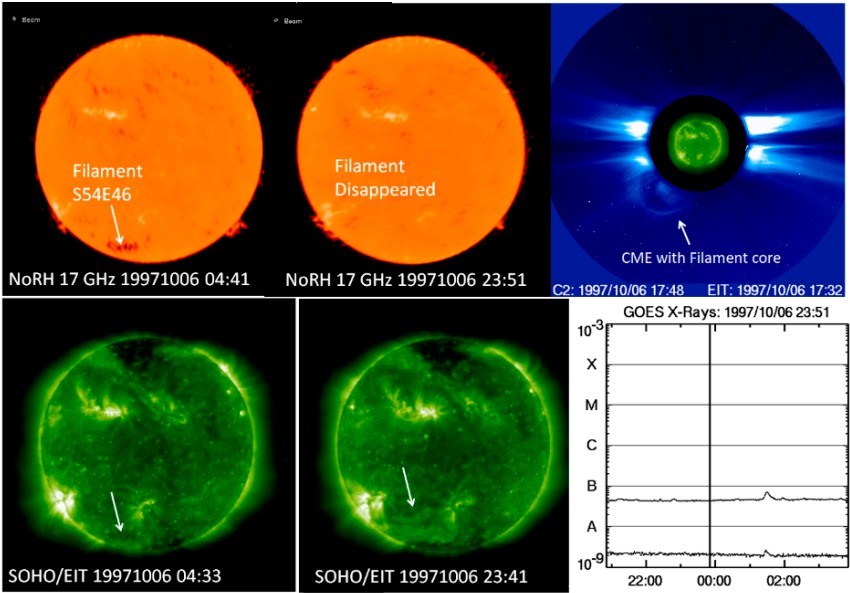

**Figure 6.** (**top row**) CME–flare relationship, illustrated using the 6 October 1997 filament eruption event. The Nobeyama Radioheliograph (NoRH) image at 04:41 UT shows the filament located in the southeast quadrant (S54E46), while the image at 23:51 UT shows the filament has disappeared and ended up as the bright core of the SOHO/LASCO CME. (**bottom row**) The filament is also observed in the 04:33 UT EUV image obtained by SOHO/EIT at 195 Å. After the filament disappeared, a post-eruption arcade formed in the vicinity of the filament's pre-eruption location. The PEA is so faint that it cannot be discerned in the GOES light curve.

Figure 7 shows an active region eruption on 4 August 2011 imaged by the Nobeyama Radioheliograph and the Solar Dynamic Observatory's (SDO's) Atmospheric Imaging Assembly (AIA) at 193 Å. The bright spot in the pre-eruption image at 03:20 UT is the sunspot in the active region emitting gyro-resonance emission. The active region contains a filament, which erupts. The image at 04:20 UT shows a large brightening, which is the PEA near the sunspot and the eruptive filament. This eruption results in a fast halo CME with an average speed of 1315 km/s and a large deceleration of −41.1 km s$^{-2}$ in the LASCO FOV. To see the early morphology, we have shown the view from the inner STEREO coronagraph COR1, located west of the Sun–Earth line (W101). The CME can be seen above the northwest limb of the Sun with a small filament core. The GOES light curve shows a major flare with a peak X-ray intensity of M9.3. In comparing this eruption with the one on 6 October 1997 (Figure 6), we see similarities in various aspects, except for the higher magnitudes of various parameters in the active region eruption. The GOES X-ray intensity is higher by more than three orders of magnitude. CMEs also have similar morphology, but the speeds are very different. While the kinematics and energetics of CMEs may differ quantitatively, the basic mechanism of eruption seems to be the same in the two cases. The primary difference is therefore the soft X-ray flare intensity.

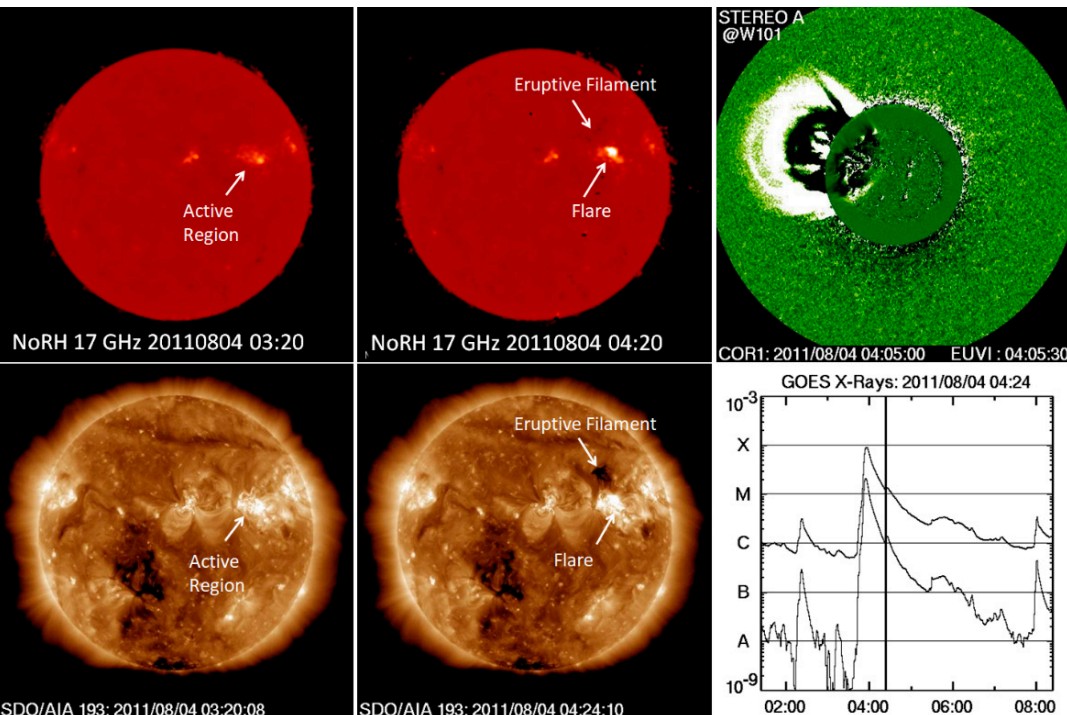

**Figure 7.** Information similar to that in Figure 6, but for the 4 August 2011 eruptive event involving a major flare. (**top row**) The Nobeyama Radioheliograph (NoRH) image at 03:20 UT shows the active region NOAA 11261 (N16W51) before the eruption. The 04:20 UT image shows the flare brightening in microwaves and the eruptive filament from the source active region. The associated CME is shown in a STEREO/COR1 image with a superposed EUV image (**top right**) showing the disturbance on the disk. (**bottom row**) The SDO/AIA images in the lower panels show the flare and filament eruption from the active region. The bright, compact PEA (marked "Flare") shows up as an M9.3 flare in the GOES light curve.

There can also be large differences in the mass emission in eruptions that have similar flare sizes. Figure 8 shows two soft X-ray flares with an X-ray flare size of X1.5. The flare on 9 March 2011 is a confined flare (no mass emission), while the eruptive flare on 2006 December 14 is accompanied by a CME that has an EUV disturbance on the solar disk and a shock in the corona. Confined flares are close to the neutral line compared to the eruptive flares [61,62]. Since flares get their energy from nonthermal particles accelerated in the corona, both types of flares involve particle acceleration, but in confined flares, these particles do not escape from the Sun (no metric radio bursts or energetic particle events in space). However, these particles do precipitate to produce hard X-ray bursts and get trapped in closed-field lines to produce microwave bursts [63]. In many cases, a series of confined flares are followed by an eruptive flare, suggesting that the confined flares might facilitate the occurrence of eruptive flares.

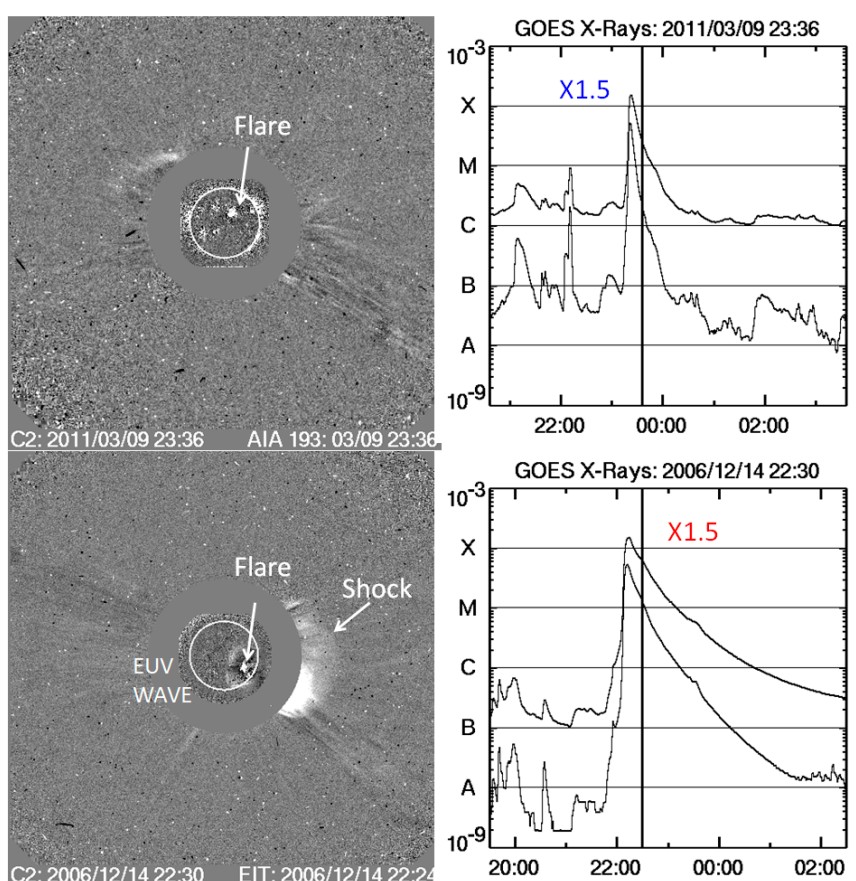

**Figure 8.** A confined (**top**) and an eruptive flare (**bottom**). The confined flare is not associated with mass motion—just electromagnetic emission (in EUV image taken by SDO/AIA). The eruptive flare has surrounding disturbances in EUV, including (from SOHO/EIT) a CME and its shock (from SOHO/LASCO). The GOES soft X-ray light curves in the right side panels show that flare intensities are very similar (X1.5). The vertical dark lines mark the time when the images on the left side panels were obtained.

## 4. CMEs and Radio Bursts

Radio bursts from the Sun have been known since the early 1940s (e.g., [64]). Nonthermal electrons accelerated during solar eruptions and other small-scale energy releases are responsible for solar radio bursts (see Figure 9). The burst types differ from one another depending on the acceleration site and the magnetic structure that carries the accelerated electrons. Type I bursts are associated with active region evolution, involving interchange reconnection between active region field lines and the neighboring open-field lines [65–68]. Type II bursts are due to electrons accelerated in the front of CME-driven shocks [69]. Type III bursts are due to electrons accelerated in a reconnection region with access to open magnetic field lines [70]. Electrons accelerated at the flare site produce stationary and moving type IV bursts when they get trapped in the PEA field lines and the CME flux rope, respectively [71]. Finally, type V bursts are a variant of type III bursts. Type III storms are the low-frequency extension of metric type I bursts, the transition happening at around 30 MHz. The individual bursts in the type III storm last much shorter than the regular type III bursts. Most of these bursts are produced by the plasma emission mechanism, involving the generation of Langmuir waves by beam–plasma instability and their conversion into electromagnetic emission [72]. Radio emission also occurs at microwave frequencies up to THz due to higher-energy electrons. These occur when high-energy electrons accelerated at the flare site are injected into flare loops where they get trapped and produce gyro-synchrotron emission.

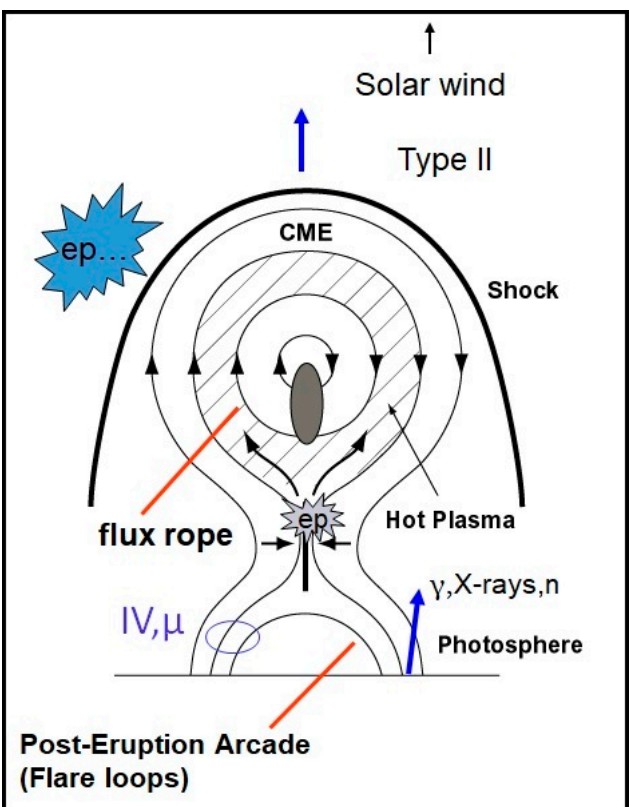

**Figure 9.** Schematic showing various aspects of a large solar eruption and particle acceleration. Two sites of particle acceleration are indicated by ep . . . , one being the flare reconnection site underneath the flux rope and the other in the shock front at the leading edge of the flux rope. The gray ellipse at the core of the CME flux rope is the eruptive prominence. Heated plasma and accelerated particles from the flare site enter the flux rope. Accelerated electrons trapped in the flux rope cause moving type IV bursts. Accelerated particles from the flare site also flow down towards the Sun causing hard X-rays (by electrons), gamma rays (by protons), and neutron emission (due to proton interaction with the chromosphere). Sunward electrons trapped in flare loops produce microwave bursts and stationary type IV bursts. Energy deposited in the chromosphere by the flare particles results in chromospheric evaporation making the flare loops hot and emit soft X-rays.

Radio bursts occurring at frequencies below the ionospheric cutoff (~15 MHz) are indicators of disturbances propagating far into the IP medium (see e.g., [73]). These are type II, type III, and type IV bursts (during eruptions) in addition to type III storms (outside of eruptions). Figure 10 identifies various low-frequency (<14 MHz) radio bursts during the 15 January 2005 eruption, around 6 UT, and illustrates how the radio bursts are related to the flare and CME during the eruption. A type III storm is in progress since the previous day and abruptly ends with the appearance of a regular type III burst, which marks the onset of the eruption. The eruption disrupts the storm, which reestablishes itself about 10 h later [74]. The eruption type III burst starts at 06:07 UT and lasts until ~ 06:40 UT. The type IV burst starts at ~06:10 UT and lasts until ~08:30 UT, extending down to ~8 MHz. The type II burst is somewhat complex with a brief fundamental–harmonic pair (at 3 and 6 MHz) around 06:47 UT and an intense one starting at 2 MHz at 06:30 UT. Examination of the corresponding ground-based observations indicate that the brief pair is a continuation of a high-frequency type II burst starting at 05:59 UT. The two sets of type II bursts are understood as the emission coming from the shock flanks (high frequencies) and nose (low frequencies). The low-frequency type IV burst is a continuation of the metric type IV burst that starts at frequencies >200 MHz. The low-frequency type III burst starts ~10 min after the start of the soft X-ray flare (05:54 UT), which peaks at 06:38 UT and ends at 07:16 UT.

Thus, the type III burst corresponds to the flare impulsive phase when most nonthermal particles are accelerated at the flare site.

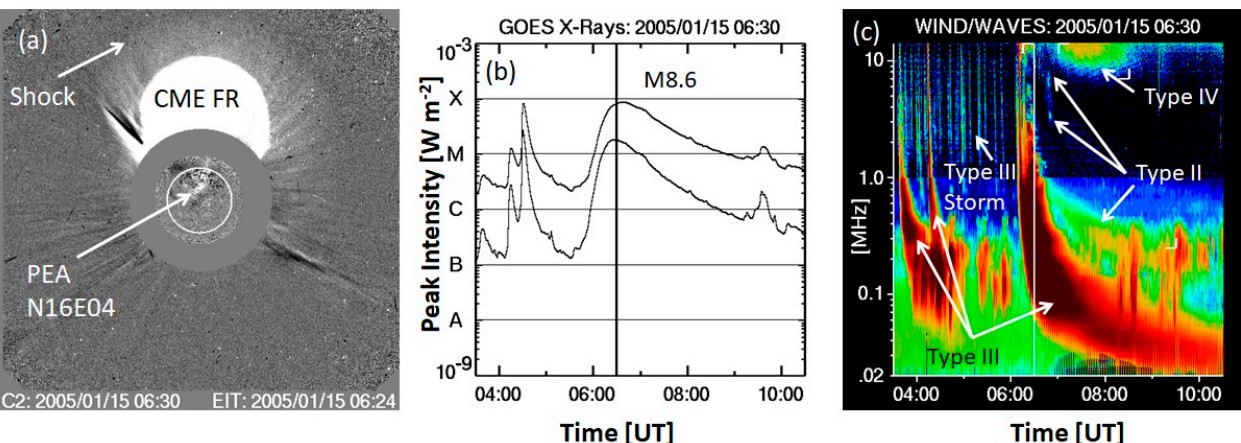

**Figure 10.** A solar eruptive event and the associated SOHO/LASCO CME (**a**), GOES soft X-ray flare of size M8.6 (**b**), and the radio bursts from the Radio and Plasma Waves experiment (WAVES) on board the Wind spacecraft (**c**). The shock in the CME image is responsible for the type II burst. The type IV burst is from the PEA. The large number of short duration bursts constitute the type III storm in (**c**). The main eruption type III burst (after 06:00 UT) and two earlier isolated type III bursts are indicated. The storm disappears after the eruption type III burst. The vertical line in the GOES plot shows the time of the CME image.

Type II, Type III, and type IV bursts occurring at frequencies below the ionospheric cutoff (~15 MHz) are indicative of large-scale eruptions, as in Figure 10, and hence are highly relevant to space weather. Of these, type II bursts are due to shocks propagating away from the Sun, and the shock formation is indicated by the onset of metric type II bursts. Type II bursts also point to the small fraction of fast and wide CMEs relevant for space weather [75]. By tracking the type II bursts to tens of kHz, it is possible to predict their arrival time at Earth [76].

Some high-frequency bursts have been found to have direct effect on global positioning system receivers [77]. On 6 December 2006, a radio burst occurred with an unprecedented intensity of $>10^6$ solar flux units (sfu). The radio burst most severely affected the civilian dual frequency GPS receivers in the Sun-lit hemisphere. Typically, signals from four members of the global navigation satellite system (GNSS) are needed to be in view for a receiver to compute positioning. During the microwave burst, the number of satellites that could be tracked fell below four, and hence the positioning accuracy degraded significantly or was not even possible using the system for tens of minutes. A similar intense microwave burst occurred in SC 24 on 4 November 2015, with an intensity of ~$10^5$ sfu. During this event, signals from secondary air traffic control radars in Sweden, Norway, and Belgium became severely disturbed when the antennas were pointed in the direction of the Sun. Examination of the radio dynamic spectra reveals that solar radio flux dramatically increased near the L band at which the GNSS and radar signals are employed. There is also a lot of variability in the intensity of neighboring frequencies. Cliver et al. [78] have concluded that such high-intensity bursts belong to the "dragon king" type of events, in that the radio emission mechanism is different from that of regular events. The dragon king events are due to a coherent radio emission mechanism such as electron cyclotron maser, as opposed to gyro-synchrotron emission for regular bursts.

X-ray photons during solar flares increase the ionization in the D and E layers of the ionosphere, thereby changing the conductivity. One of the consequences is the impact on the very low frequency (VLF) waves that bounce off from the bottom of the ionosphere. The amplitude and phase of the VLF waves are altered by the flare-induced changes in the ionospheric conductivity. By monitoring the VLF waves, one can detect solar flares of

B5-class and above and the associated ionospheric disturbances [79]. Intense solar flares also cause the so-called magnetic crochet, which is a minor disturbance of Earth's magnetic field [80]. The flare intensity needs to be about two orders of magnitude higher to cause crochets than that causing sudden ionospheric disturbances [81].

## 5. Solar Connection to Geomagnetic Storms

It was recognized a long time ago that geomagnetic disturbances are intimately related to the southward IP magnetic field [82,83]. When the southward magnetic field component of an IP structure such as an ICME reconnects with the Earth's field in the magnetosphere, a geomagnetic storm ensues. Following the dayside reconnection, a nightside reconnection occurs and particles are injected into the magnetosphere, enhancing the ring current, which affects Earth's magnetic field at ground level [82]. The storm strength is measured by an index such as Dst, which is an average deviation of Earth's horizontal magnetic field measured at four low-latitude magnetometer stations [84] (https://wdc.kugi.kyoto-u.ac.jp/dstdir/dst2/onDstindex.html, accessed on 1 October 2022). While the southward magnetic field is necessary for a storm, the storm strength is determined by additional solar wind parameters such as the speed and dynamic pressure [85–91].

An IP structure that causes a geomagnetic storm is said to be geoeffective. In the undisturbed solar wind, the IP magnetic field is in the ecliptic plane and hence does not have an out-of-the-ecliptic component (Bz), except for Alfven waves. The CME connection to geomagnetic storms comes from the fact that the IP manifestation of CMEs such as magnetic clouds [92,93] possess significant Bz, which causes a geomagnetic storm when negative (southward) [94–96]. Sheath regions behind ICMEs often contain Bz < 0 and cause geomagnetic storms [97,98].

Coronal holes are regions of the corona where the density is low and the photospheric magnetic field underlying them is predominantly unipolar, indicating open magnetic flux (see [99] for a review). Plasma is free to escape along the open-field lines, observed as an HSS. The speed of HSS observed at 1 au has been found to depend on the coronal hole area, expansion factor of the magnetic field, and the photospheric magnetic field strength [100–105]. Coronal holes are generally of limited spatial extent, so an HSS typically presses against a preceding slower wind, forming a SIR. When a SIR is observed for more than one solar rotation, it is called a corotating interaction region (CIR). A CIR/SIR is identified based on the increase in density, temperature, and magnetic field strength during the positive gradient of the solar wind speed [106–110]. The occurrence rate of SIRs is solar-cycle dependent, with a majority of them occurring during the declining phase of a solar cycle (see e.g., [111] and references therein). The solar cycle variation of CIRs reflect the occurrence pattern of coronal holes on the Sun at low and high latitudes [103,112]. SIRs possess enhanced density, dynamic pressure, temperature, magnetic field strength, and speed compared to the preceding solar wind. Many of these are important in causing geomagnetic storms [113]. Table 1 compares the SIR parameters with the corresponding ones in the solar wind (from [114]).

**Table 1.** Properties CIR and solar wind parameters.

| Parameter | CIR | | Solar Wind | Ratio |
|---|---|---|---|---|
| | Range | Mean | | |
| Density [cm$^{-3}$] | 2.4–81.0 | 29.3 | 6.7 | 2.93 |
| Dynamic pressure [nPa] | 1.4–57.2 | 10.5 | 2.3 | 4.57 |
| Temperature [$10^5$ K] | 0.97–26.35 | 4.91 | 1.02 | 4.81 |
| Magnetic field [nT] | 4.6–44.9 | 14.8 | 5.5 | 2.69 |
| Duration [hr] | 2.75–82.10 | 26.47 | — | — |
| Extent [au] | 0.03–0.98 | 0.31 | — | — |

### 5.1. CMEs and Geomagnetic Storms

CMEs are thus a major source of southward IMF (Bz < 0) owing to their flux rope nature and shock-driving capability ([98] depending on the location(s) of Bz < 0 within the ICME, the storm can start anytime from sheath arrival to the arrival of the back of the ICME [115]). One can think of the following geoeffectiveness scenarios depending on the location of Bz < 0 interval within the overall structure of ICMEs: (1) both sheath and cloud are geoeffective, (2) sheath alone is geoeffective, and (3) cloud alone is geoeffective, and (4) neither sheath nor cloud is geoeffective. When both the sheath and cloud portions are geoeffective, the Dst (disturbance storm time) profile can be complex, leading to multistep storms [116,117].

Figure 11 shows a shock-driving CME and a schematic of its IP manifestation. The flux rope is a bundle of helical field lines that are rooted on either side of the neutral line in the source region on the Sun. In the cross-sectional plane, the field lines appear circular, with the front and back field lines pointing in opposite directions. In reality, the cross-section can be elliptical or heavily deformed due to interaction with the ambient solar wind. If the MC arrives at Earth in the flux rope configuration shown Figure 11, it will be termed as south–north (SN) MC. If the rotation is reversed, it represents a north–south (NS) MC, indicating that the leading edge now has a north-pointing magnetic field component. The NS and SN MCs are known as bipolar, as opposed to the unipolar MCs in which the axial field is in the north–south direction, while the field rotates in the east–west direction. Unipolar MCs are called fully-north (FN) or fully-south (FS) to indicate that the axial field points to the north or south, respectively. More details can be found in [118–124]. The onset of a geomagnetic storm can be delayed with respect to the arrival time of the MC, depending on the MC type and the presence of a sheath [115]. The SN, FS, and NS type MCs have average delays of about 6, 9, and 19 h, respectively, from the cloud arrival to the Dst minimum time. The SN and NS type MCs have a similar storm strength (Dst ~ −107 and −104 nT, respectively), and the FS MCs result in stronger storms (average ~ −125 nT). Sheath storms attain their peak strengths about 3–4 h before the MC arrival because the shock arrives earlier. When the back of MCs contain high-density material due to filaments [36,91] or when compressed by a CIR [125], the geoeffectiveness can be enhanced.

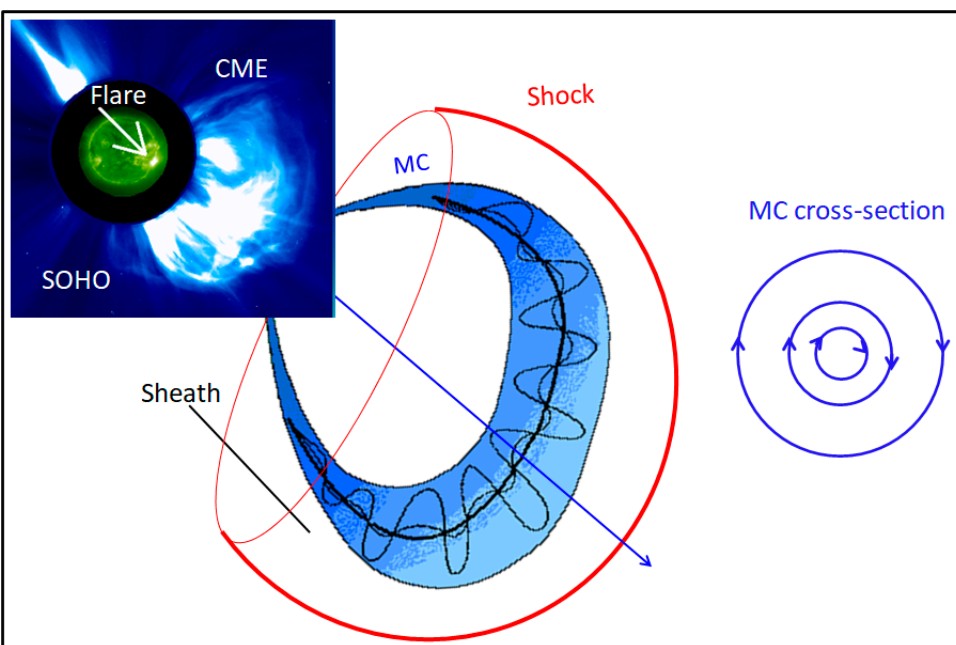

**Figure 11.** Illustration of the CME structure responsible for geomagnetic storms. The SOHO/LASCO image of a CME with the flare location shown in the superposed SOHO/EIT image (**top left**). The

CME is observed in the IP medium as a magnetic cloud (MC) driving a shock. The blue straight arrow points to the direction of motion of the flux rope with the shock. The blue concentric circles represent the cross-section of the MC flux rope, with the arrows indicating direction of field lines in a direction perpendicular to the flux rope axis (**right**). The downward (**upward**) arrows denote field direction pointing southward (**northward**) in the IP medium. When the field points southward, a geomagnetic storm ensues. When the flux rope axis is in the ecliptic plane, the azimuthal field becomes the Bz component. When the axis is highly inclined with respect to the ecliptic, the axial field becomes Bz. When shock-driving (see Figure 1), the sheath ahead of the flux rope contains Bz. CIRs can also be a source of Bz because they amplify the solar wind Alfven waves in the compression region.

Examples of a double-dip storm and a sheath storm are shown in Figure 12. The underlying CMEs occurred on 28 and 29 October 2003 at the Sun [126]. The two CMEs are fast halo CMEs (speed > 2000 km/s) that ended up as dissimilar ICMEs. The first ICME has Bz < 0 in the sheath, followed by a large Bz < 0 in the cloud. In the second MC, the sheath has a large Bz < 0 with mostly Bz > 0 in the cloud. The reason for the different appearance of MCs is that they originate from different neutral lines in the source active region. The neutral line/filament is a first good indicator of the expected orientation of the flux rope axis in the IP medium [127].

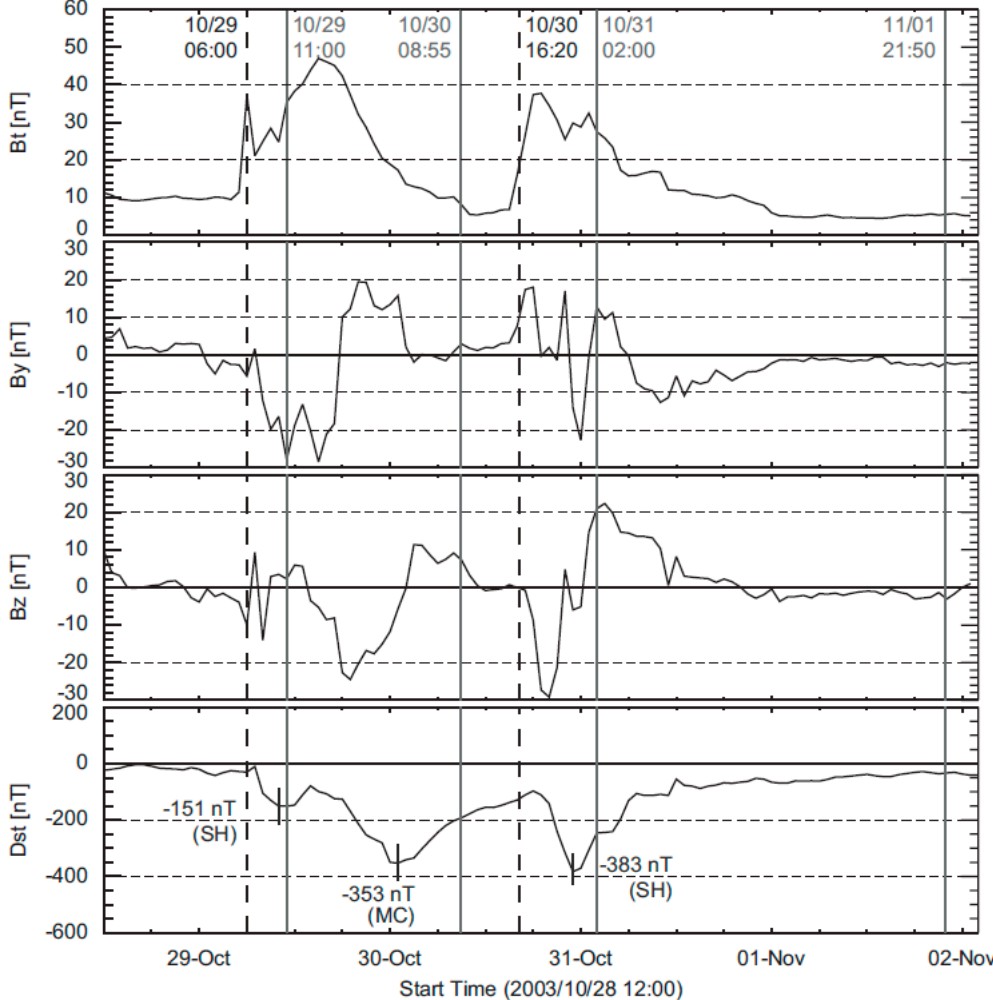

**Figure 12.** Two super-intense magnetic storms from the Halloween 2003 period, one caused by an FS MC, the other caused by a sheath. The underlying CMEs originated from the same active region, one day apart (from [115]).

The speed and magnetic content of ICMEs are ultimately connected to the free energy in the source magnetic region on the Sun. One of the parameters that is readily measured during an eruption is the total reconnected flux, which is highly correlated with the CME speed [128,129], CME kinetic energy, and flare fluence [130]. By combining the reconnected flux and the geometrical parameters of the CME obtained from flux rope fit to white-light images, it has been shown that the axial field strength near the Sun is correlated with the CME speed [130]. Such a relationship was obtained in MCs at 1 au [122,131]. These studies suggest that faster CMEs are likely to cause stronger geomagnetic disturbances when Bz < 0.

Figure 13 shows three observational facts about CMEs causing intense (Dst $\leq -100$ nT) geomagnetic storms: (i) The CMEs are two times faster than typical CMEs (955 km/s vs. 416 km/s). (ii) Two thirds of the storm-causing CMEs are halos; the average width of non-halo CMEs is 175°, much larger than the typical width of CMEs, viz, 38°. (iii) The heliographic locations of the storm-causing CMEs are concentrated near the solar disk center, especially the ones causing more intense storms. The geoeffectiveness of CMEs decreases as a function of the central meridian distance [132]. (i) and (ii) imply that the CMEs are very energetic (wider CMEs are more massive, and hence the kinetic energy is high—see [55]). (iii) implies that CMEs heading directly toward Earth are more impactful in causing geomagnetic storms. Note that almost all storms with intensities < $-200$ nT are within ±30° longitude. This was recognized long time ago by H. W. Newton [133], who found that the locations of the flares associated with great storms are close to the central meridian, with a slight bias to the Western Hemisphere (see also [1] for details). The slight western bias has been demonstrated using CME data by [134]. The western bias is related to the fact that CMEs are deflected slightly eastward due to solar rotation [135]. The source locations are also distributed around N15 and S15 latitudes, which correspond to the active region belt. Active regions possess the highest levels of magnetic energy needed to power these energetic CMEs. Another implication of the source locations close to the disk center is that the CME speeds are underestimated because of projection effects. If we apply the empirical relationship V3D = 1.1 Vsky + 156 ([136]), we see that for Vsky = 955 km/s, the average 3D speed (V3D) of storm-producing CMEs becomes 1207 km/s. Occasionally, CMEs originating close to the limb also cause intense storms. There are five limb halos (CMD $\geq 60°$) in Figure 13c that produced intense storms. These CMEs are geoeffective because they are very energetic and their sheath with significant Bz < 0 component is intercepted by Earth. One of these storms (Dst = $-288$ nT) is due to the 4 April 2000 west limb CME with a sky-plane speed of 1188 km/s [137–139]. The deprojected speed is ~1450 km/s.

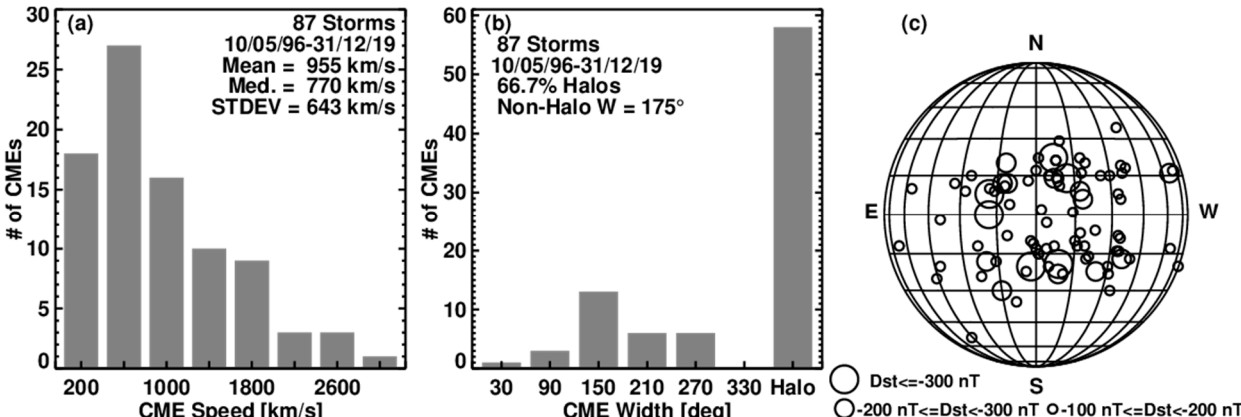

**Figure 13.** # means Number in Figure 13a,b. Distributions of speed (**a**), width (**b**), and source locations (**c**) of CMEs that resulted in intense geomagnetic storms (Dst $\leq -100$ nT), observed over a period of ~23 years. In (**c**) three storm intensity levels are distinguished by the size of circles. Geoeffective CMEs are fast and wide, and they originate from close to the solar disk center.

### 5.2. Coronal Holes and Geomagnetic Storms

SIRs often act like CMEs in their Earth impact, causing geomagnetic storms of intensities up to ~150 nT [115,140,141]. Figure 14 shows a low-latitude coronal hole that resulted in a HSS with a speed of ~750 km/s. The HSS caused a CIR in which the density attained a peak value of ~30 cm$^{-3}$. The Bz component was relatively large (−20 nT) and resulted in a geomagnetic storm with Dst = −119 nT. Investigating the geoeffectiveness of 866 SIRs during 1995–2016 [141], it was found that about half of them (52%) caused some level of geomagnetic storms (Dst $\leq$ −30 nT). The number of SIRs causing geomagnetic storms rapidly decreases with storm intensity: minor (−50 nT < Dst $\leq$ −30 nT), moderate (−100 nT < Dst $\leq$ −50 nT), and intense storms (Dst < −100 nT) are caused by 240 (28%), 187 (22%), and 26 (3%), respectively. Although weak, the SIR storms occur more frequently than the ICME storms and hence are very important for space weather (e.g., [142]).

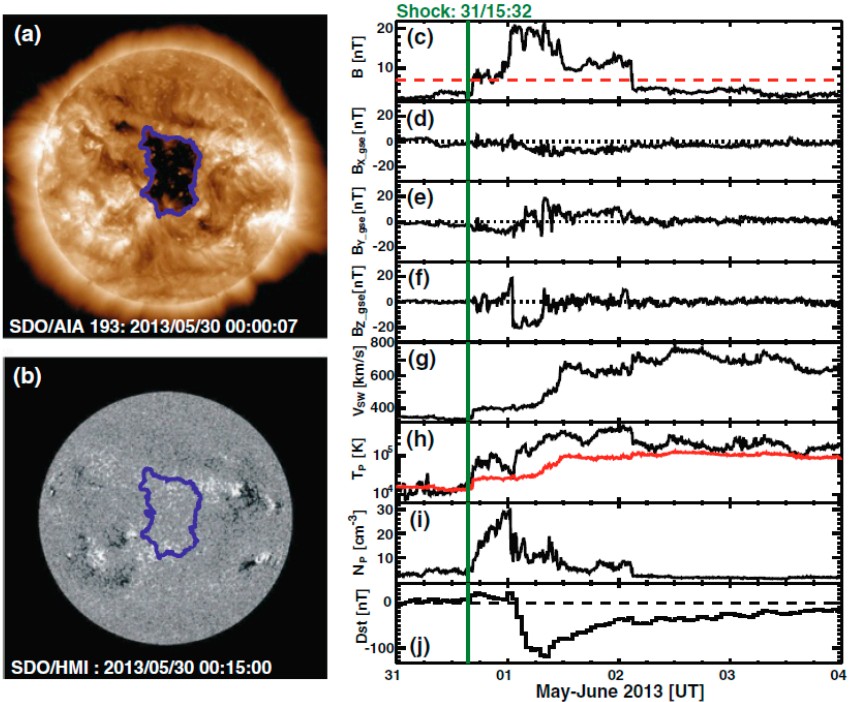

**Figure 14.** A low-latitude coronal hole observed by SDO/AIA at 193 Å (**a**), the underlying photospheric magnetic field strength from SDO/HMI (**b**), and solar wind parameters inaround the CIR with a leading shock (**c**) (from Gopalswamy et al. 2015). The coronal hole was near the central meridian on 30 May 2013 and resulted in an intense storm (−119 nT) on 1 June. Magnetic field components Bx, By, and Bz are shown in panels (**d**–**f**), respectively. The three indicators of a stream interaction region are the increase in solar wind speed (**g**), increasing temperature (**h**), and density peak (**i**). The Dst index (**j**) shows an intense storm (Dst < −100 nT).

Recent investigations have shown that the space weather due to CIRs are milder in SC 24 [114,143–146]. Gopalswamy et al. [144] reported that the number of intense geomagnetic storms caused by CIRs has dropped by 75% in SC 24. Grandin et al. [146] report that stream SIR/HSSs are 20–40% less geoeffective during cycle 24 than during SC23. The speed and magnetic field strength in cycle 24 are smaller than the corresponding values in cycle 23 and hence the weaker geomagnetic storms.

## 6. Solar Eruptions and SEPs

Solar energetic particles (SEPs) are nonthermal electrons and ions that have energies well above the thermal plasma particles. Energetic ions can be accelerated up to multi-GeV energies during solar eruptions. The energetic electrons and ions are detected directly by particle detectors in space. They are also inferred from the electromagnetic emissions they

produce while interacting with the ambient medium. SEPs were first observed in the early 1940s using ionization chambers at the ground level [147]. The SEP events detected at Earth's surface are called ground-level enhancement (GLE) events. GLE events are now detected using neutron monitors or muon telescopes that detect secondary neutrons and muons caused by primary SEPs. The events reported by Forbush [147] were associated with intense H-alpha flares on the Sun; thus, flares became known as the accelerator of SEPs. Type II solar radio bursts were first detected in 1947 [148,149] and have been attributed to a fast-mode MHD shock [150]. Type II bursts are caused by nonthermal electrons accelerated at the shock via the plasma emission mechanism [72]. Lin et al. [151] found that energetic proton events occur in major eruptions accompanied by type II and type IV radio bursts, intense X-ray and microwave emission, and relativistic electrons. Rao et al. [152] suggested that energetic storm particle (ESP) events are accelerated by IP shocks when they are at Earth. Kahler et al. [153] attributed the shocks to CMEs, and hence the shock paradigm for SEPs became firmly established [154–156].

The importance of CMEs in the occurrence of SEP events is illustrated in Figure 15. These are CMEs associated with large SEP events, defined as those with proton intensity expressed in particle flux units (pfu = 1 particle cm$^{-2}$ s$^{-1}$ sr$^{-1}$) exceeding 10 in the >10 MeV energy channel. Such events have been determined to have important space weather consequences by the NOAA. The CME speeds range from ~600 km/s to >3000 km/s, with an average speed of ~1500 km/s. The speed is clearly much larger than that of an average CME. A vast majority (>80%) of SEP-associated CMEs are halos, indicating that such CMEs are very wide, further confirmed by the average width (~180°) of the non-halo CMEs. The distribution of CME source locations on the Sun heavily favors the Western Hemisphere because the accelerated particles propagate along magnetic field lines that have a Parker spiral configuration. The nominal connection angle is ~W58 for a background solar wind speed of ~ 400 km/s. While most of the high-intensity events are from the Western Hemisphere, there are some events originating from the Eastern Hemisphere, and even from the east limb. The east-limb events are of low intensity (≤100 pfu), but the CME speeds are extraordinarily high. For example, the average CME speed of the seven east-limb SEP events in Table 2 is ~1956 km/s, while the average SEP intensity (Ip) detected by GOES is only 44 pfu. The low intensity is a consequence of the poor connectivity—Earth is connected to the extreme west flank of the CME shock. It must be noted that a spacecraft behind the east limb is well-connected to such east-limb events and hence would observe an intense particle event. In Table 2, the last three events are observed by STEREO Behind (STB), which is better connected to the solar source, and hence the SEP intensity (Ip) is higher by 1–2 orders of magnitude.

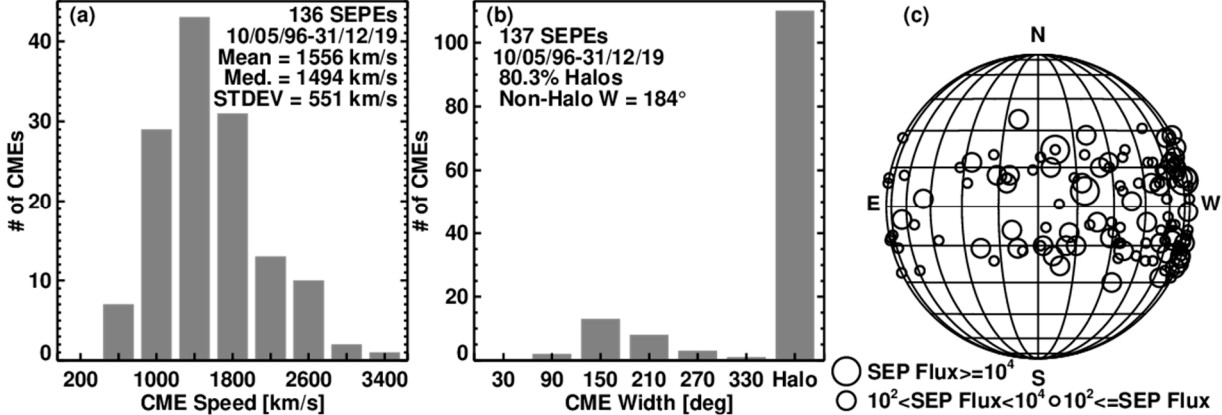

**Figure 15.** # means Number in Figure 15a,b. Same as Figure 13, but for CMEs associated with large SEP events: (**a**) speed, (**b**) width, and (**c**) source locations Three intensity levels of >10 MeV protons are distinguished by the size of the data points in (**c**).

**Table 2.** CME speeds and SEP intensities of east-limb events observed by GOES and STB.

| CME Date | Time (UT) | Speed (km/s) | Location | Ip (GOES) | Ip (STB) |
|---|---|---|---|---|---|
| 28 December 2001 | 20:30 | 2216 | S26E90 | 76 | — |
| 8 January 2002 | 17:54 | 1794 | >NE90 | 28 | — |
| 27 July 2005 | 04:54 | 1787 | N11E90 | 41 | — |
| 22 September 2011 | 10:48 | 1905 | N09E89 | 35 | 5000 |
| 21 June 2013 | 03:12 | 1900 | S16E73 | 14 | 100 |
| 25 Febraury 2014 | 01:25 | 2147 | S12E82 | 24 | 300 |

SEPs have several space weather consequences in the heliosphere. Interplanetary spacecraft and satellites in Earth's orbit can be directly affected by SEPs. The powerful eruption on 21 April 2002 (see Figure 16) involved a major flare and an ultrafast (~2400 km/s) CME that resulted in a very intense and hard-spectrum SEP event [157]. Earth was well-connected to the eruption source and hence was immersed in the particle radiation for several days. The Nozomi spacecraft, on its journey to Mars, was in the vicinity of Earth around this time and hence was impacted by the particle radiation. Nozomi's communications and power systems were damaged, causing the hydrazine to freeze in the spacecraft's attitude control system [158]. This led to a series of issues that ended the mission in December 2003, without reaching Mars.

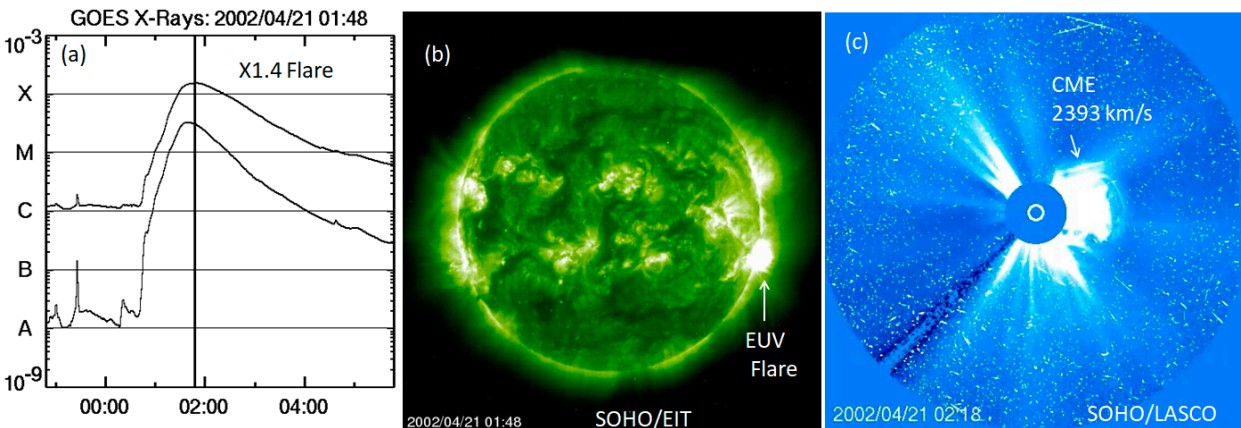

**Figure 16.** The 2002 April 21 eruption: (**a**) GOES soft X-ray flare, (**b**) the flare imaged by SOHO/EIT, and (**c**) the CME imaged by SOHO/LSACO. The dots and streaks in the CME image are due to energetic particles from the CME impacting the SOHO/LASCO detector and are referred to as a "snowstorm".

Another well-known loss to particle radiation is the Martian Radiation Environment Experiment (MARIE) on the Mars Odyssey mission. MARIE was a dedicated energetic charged particle spectrometer intended to make measurements of the particle radiation levels on the way to Mars, and in orbit it was intended to aid designers of future missions involving human explorers. When the Halloween SEP event on 28 October 2003 started, Mars Odyssey went into a safe mode. When the spacecraft came out of the safe mode, MARIE was found non-responsive, and all attempts to revive the instruments were unsuccessful; consequently, the instrument was abandoned [159].

## 7. Space Weather Events and Spacecraft Anomalies

SEPs are known to impact satellites in Earth's orbit. For example, when satellite solar panels are directly exposed to energetic protons, the current generated by the panels decreases significantly and permanently. Marvin and Gorney [160] reported that two large SEP events, which occurred on 29 September 1989 and October 19, decreased the expected current from GOES-7 solar arrays by ~5–10%. Iucci et al. [161] performed a statistical analysis of a large number of spacecraft anomalies in different Earth orbits. They found

that spacecraft in high-altitude, high-inclination (HH) orbits exhibited higher frequencies of spacecraft anomalies compared to those at lower altitudes and inclinations (LL, LH), as summarized in Figure 17. We see that the anomaly frequency is the highest for spacecraft in high-altitude, high-inclination (HH) orbits. These are the orbits in which most of the GNSS satellites are located. The anomaly frequency rapidly increases with proton flux. For example, for HH orbits, the anomaly frequency increases by an order of magnitude when the proton flux increases from 100 to 1000 pfu. The probability of an anomaly for HH orbit is significantly higher for higher proton flux and proton energy. Finally, the anomaly frequency peaks typically 4–5 days after the onset of the proton event.

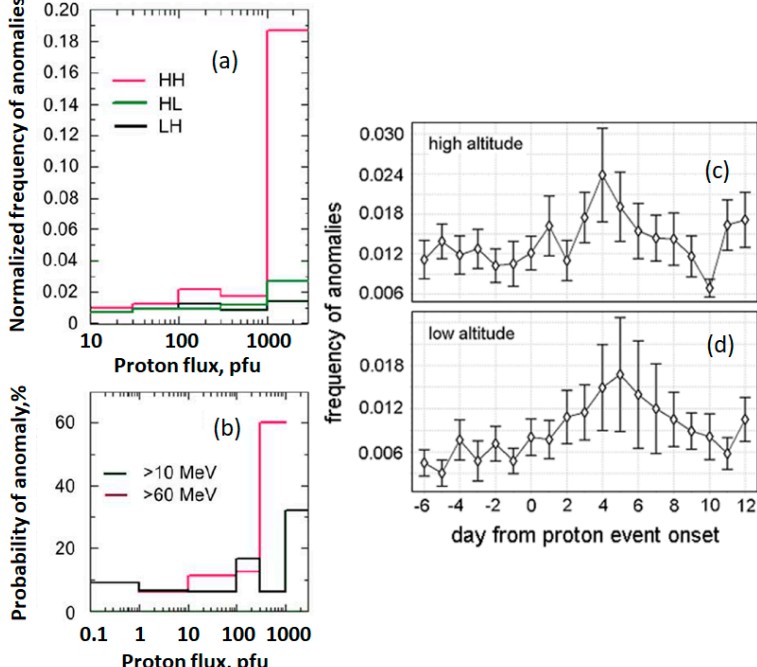

**Figure 17.** (**a**) The normalized frequency of spacecraft anomalies (averaged over the first 2 days of proton enhancements) as a function of maximal proton flux at energies >10 MeV from IMP8 for various Earth orbits: high altitude with high inclination (HH, red), high altitude with low inclination (HL, green), and low altitude with high inclination (LH, black). (**b**) The probability of anomaly as a function of proton flux in two different energy channels (>10 MeV in black and >60 MeV in red). Particle data in (**a**,**b**) are from IMP 8. The frequency of anomalies as a function of time elapsed since the onset of a proton event for high-altitude (**c**) and low-altitude (**d**) spacecraft, irrespective of inclination (adapted from [161]).

Interestingly, the spacecraft anomalies also peak following geomagnetic storms. Figure 18 shows the anomaly frequency as a function time starting from the time of storm sudden commencement (SSC). For satellites in high latitudes and low latitudes, the anomaly frequency peaks about three and five days after SSC, respectively. The anomaly peaks roughly coincide with the time of peak relativistic electron flux following the initiation of a geomagnetic storm. The relativistic electron flux peaks earlier at lower energies because of the progressive energization of electrons by low-frequency waves generated by low-energy particles injected into the magnetosphere during storm-time substorms (see e.g., [162]). The relativistic electron flux is enhanced both during CME and CIR storms, although CIR storms elevate the flux to much higher levels than the CME storms do [163]. The relativistic electrons can be as harmful as the SEPs in causing satellite anomalies [164].

The spacecraft anomalies occur because of the interaction between spacecraft and their hazardous environment. The resulting impact depends on the energy and the type of particles involved [166]. Table 3 lists the impact on spacecraft by electrons, protons, and heavier ions of various energies and sources.

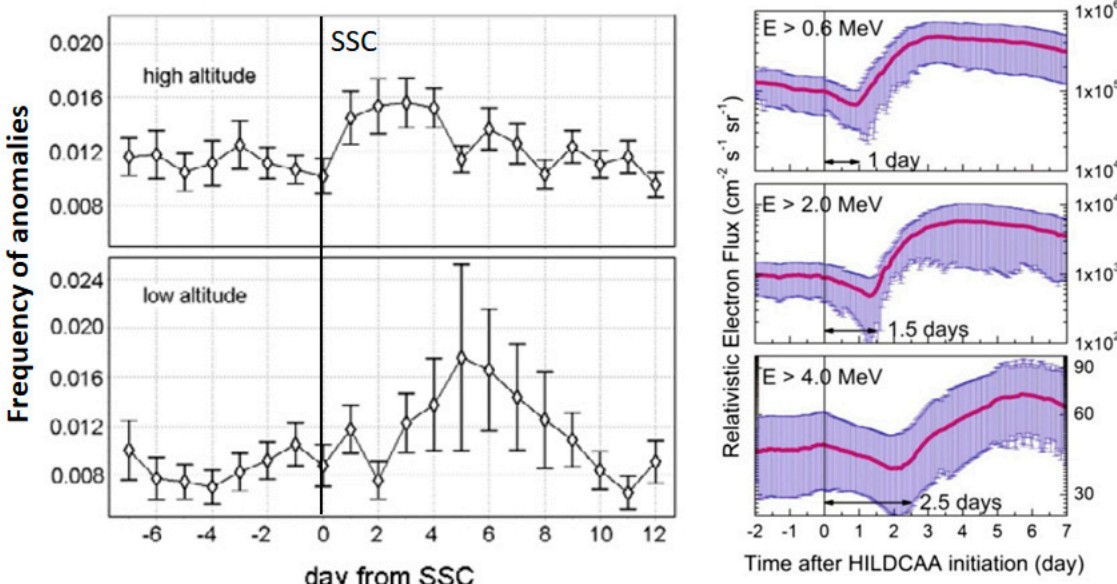

**Figure 18.** (**left**) The frequency of spacecraft anomalies as a function of time elapsed since the storm sudden commencement (SSC) (from [161]). (**right**) The progressive increase in relativistic electron flux at various energies following a geomagnetic storm due to a high-intensity, long-duration continuous auroral electrojet activities (HILDCAA) event (from [165]). The initial dip in the electron flux is due to the compression of the magnetosphere by the HSS.

**Table 3.** Particles in various energy ranges, their sources, and their effects.

| Particle Type | Energy Range | Effects | Sources |
|---|---|---|---|
| Electrons | 10–100 keV | Spacecraft charging | Trapped particles |
| Electrons | >100 keV | Deep dielectric charging, solar cell damage | Trapped particles |
| Electrons | >1 MeV | Radiation damage (ionization) | Trapped/quasi trapped |
| Protons | 0.1–1 MeV | Surface damage to materials | Trapped particles |
| Protons | 1–10 MeV | Displacement damage in solar cells | Trapped particles, ESP |
| Protons | >10 MeV | Ionization, displacement damage; sensor background | Radiation belt, SEPs, GCRs |
| Protons | >30 MeV | Damage to biological systems | Radiation belt, SEPs, GCRs |
| Protons | >50 MeV | Single event effects | Radiation belt, SEPs, GCRs |
| Ions | >10 MeV/nuc | Single event effects | SEPs, GCRs |
| Protons | >500 MeV | Single event effects, hazard to humans in polar flights and in deep space | SEPs, GCRs |

The relativistic electrons noted above are part of the highly variable Earth's outer radiation belt. The inner belt is populated mainly by energetic protons resulting from the galactic cosmic rays via the CRAND (cosmic ray albedo neutron decay) mechanism (e.g., [167]. SEPs from energetic CMEs also contribute to the inner belt (see e.g., [168]). The CRAND mechanism can also contribute to energetic electrons in the inner belt [169].

Precipitation of radiation belt particles, SEPs, and GCRs to the Earth's polar atmosphere affects the atmospheric chemistry, including ozone depletion (see [170] and references therein). SEPs penetrate the polar atmosphere to various depths depending on their energy: 1, 10, 100, and 1000 MeV particles can penetrate to the mesopause, mesosphere, stratosphere, and troposphere, respectively. The 100 MeV particles in the stratosphere can dissociate molecules to produce radicals such as HOx and NOx that react with ozone,

contributing to ozone depletion [171]. The GeV particles reach the troposphere, where the primary particles produce air shower, including secondary particles such as muons and neutrons detected by ground-based monitors (see e.g., [172] for a review). GLEs have a harder spectrum than regular SEP events and SEP events associated with filament eruption CMEs [157]. GLEs deposit their energy in Earth's polar and mid-latitude regions. Because of their higher energy, GLEs also significantly impact solar cells and star-sensor pointing systems on spacecraft. They also increase of the radiation environment on spacecraft components and transpolar aircraft.

Solar disturbances have a significant impact on terrestrial technological systems as well [173]. The impact is in the form of a geomagnetically induced current (GIC) caused by rapid variation in ionospheric currents during a shock compression of the magnetosphere (SSC), a substorm, or other fast processes [174–176]. GICs affect any large-scale conducting system at the surface of Earth such as railroads, telephone lines, pipelines, and electric power grids. Hazardous GICs have been found to be mainly associated with CMEs rather than CIRs [163,177]. GICs at high latitudes have been extensively studied (see [178–180] and references therein). GICs can be significant at mid and low latitudes as well [181,182]. SSCs are followed by geomagnetic storms. Sudden impulses represent the arrival of CME-driven shocks at the magnetosphere but are not followed by geomagnetic storms. Even during sudden impulses, GICs can increase significantly [182].

## 8. Solar Cycle Variation of Space Weather Events

Solar activity represents the appearance and dispersal of closed and open magnetic field regions on the Sun. While CMEs originate from closed-field regions, high-speed streams originate from open-field regions. Indices such as sunspot number (SSN) and the radio flux at 10.7 cm wavelength (F10.7) are typical measures of solar activity, although there are many other indicators. For example, the magnetic butterfly diagram [183] provides information on the magnetic nature of solar activity and how the low-latitude and high-latitude magnetic regions are related. Since CMEs and CIRs originate from enhanced closed- and open-field regions on the Sun, solar activity has clear relevance to space weather events. When the SSN is high (solar maximum phase), there are more closed-field regions on the Sun, and hence the probability is high for the occurrence of CMEs. Similarly, when there are many low-latitude coronal holes in the declining phase of the solar cycles, there are more HSS and the related CIRs and geomagnetic storms.

### 8.1. CME Rate–Sunspot Number Relationship

The overall solar cycle variation of the daily rate of CMEs is very similar to that of SSN in phase. However, the amplitudes of the two phenomena are different in different solar cycles (Figure 19): the CME rates are similar in SCs 23 and 24, but the SSN is much smaller in SC 24. However, fast and wide (FW) CMEs relevant for space weather are very different between the two cycles and consistent with the reduction in SSN. The number of major flares is also smaller in SC 24. The occurrence of less energetic CMEs in SC 24 is further supported by the smaller average CME speed in that cycle (Figure 19c). The rise phase of SC 25 has witnessed CME and SSN behavior similar to those in SC 24.

Figure 20 further examines the relation between the CME rate and SSN using a scatter plot. Considering all CMEs observed by SOHO/LASCO from 1996 to 2021, we see that the CME rate–SSN correlation is high (r = 0.82). There is a large scatter in the SSN range 100–150, which corresponds to the maximum phase. The scatter is drastically reduced when the data points are separated according to the solar cycle they belong to. The correlation improves significantly to r = 0.88 for SC 23 and 0.92 for SC 24. The different slopes of the regression lines are consistent with the higher amplitude in the CME rate, as shown in Figure 19. However, it must be noted that the CME identification was made by several people, so one cannot rule out the effect of subjectivity on the CME rate in cycle 23 (especially of the slow and narrow CMEs). In addition to the inter-cycle variations, the correlation shows intra-cycle variations as well (Figure 21). While the CME rate–SSN correlation

is similar in the rise and declining phases of the two cycles, it is relatively small in the maximum phases (r = 0.63 in SC 23 and 0.71 in SC 24). The reduction in the correlation has been attributed to the non-spot CME sources that are abundant during the maximum phase [184,185]. Figure 22 illustrates this using the locations of prominence eruptions and their occurrence rates obtained from the Nobeyama Radioheliograph images [186]. While the sunspots occur only at low latitudes, prominences occur at all latitudes. There is an abundance of prominence eruptions at latitudes 30–60°, with an additional population at latitudes >60° due to the rush-to-the-poles phenomena. The occurrence rate and source distribution of prominence eruptions are very similar to those of regular CMEs because the two phenomena are physically related (see [50,187] and references therein). Thus, the non-spot CME rate is not expected to correlate with SSN, resulting in the overall reduction in the CME rate–SSN correlation.

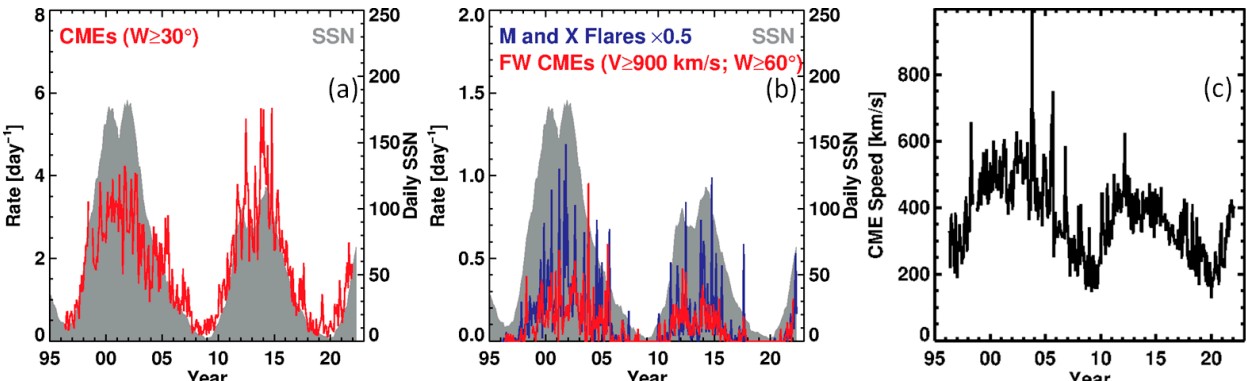

**Figure 19.** The occurrence rate of (**a**) regular (width ≥ 30°) and (**b**) fast and wide (FW) CMEs, observed by SOHO/LASCO (in red) since 1996 (superposed on the daily SSN in gray). Also shown in blue are the number of major X-ray flares (M and X class) in (**b**). (**c**) CME speed averaged over Carrington rotation periods.

### 8.2. Solar Cycle Dependence of Space Weather Consequences

The reduction of solar activity in SC 24 resulted in mild space weather during that cycle. Figure 23 shows the time variation of the number of CME-associated major geomagnetic storms (Dst < −100 nT) and large SEP events (>10 MeV proton intensity ≥ 10 pfu) since 1996. We see that the numbers dropped by 74% and 55%, respectively. When the CIR and CME storms are combined, the reduction is similar. The number of major storms due to CIRs drops from 9 to 2 (or by 78%). The reduction in the number of storms is more than that in SSN and FW CMEs. This can be attributed to the weakened state of the heliosphere in cycle 24. The reduced solar activity results in a weaker heliospheric pressure, which backreacts on CMEs, making them magnetically dilute due to the anomalous expansion. Furthermore, the SC-24 CMEs are slower on average. Since the storm strength is primarily decided by the product of CME speed and the southward IMF, reduction in both factors is responsible for the reduced number of storms. In the case of CIR storms, the reduced heliospheric magnetic field should result in reduced field strength in the compressed interface, contributing to the reduced number of geomagnetic storms. In the case of SEP events, the reduction is primarily due to the reduced number of FW CMEs. The severest reduction is in the number of GLEs: 16 in SC 23 vs. just 2 in SC 24 (i.e., an 87% drop). This has been attributed to the reduced acceleration efficiency in the weakened ambient magnetic field, so particles did not attain high energies [55,144]. An additional reason could be the presence of fewer seed particles [188].

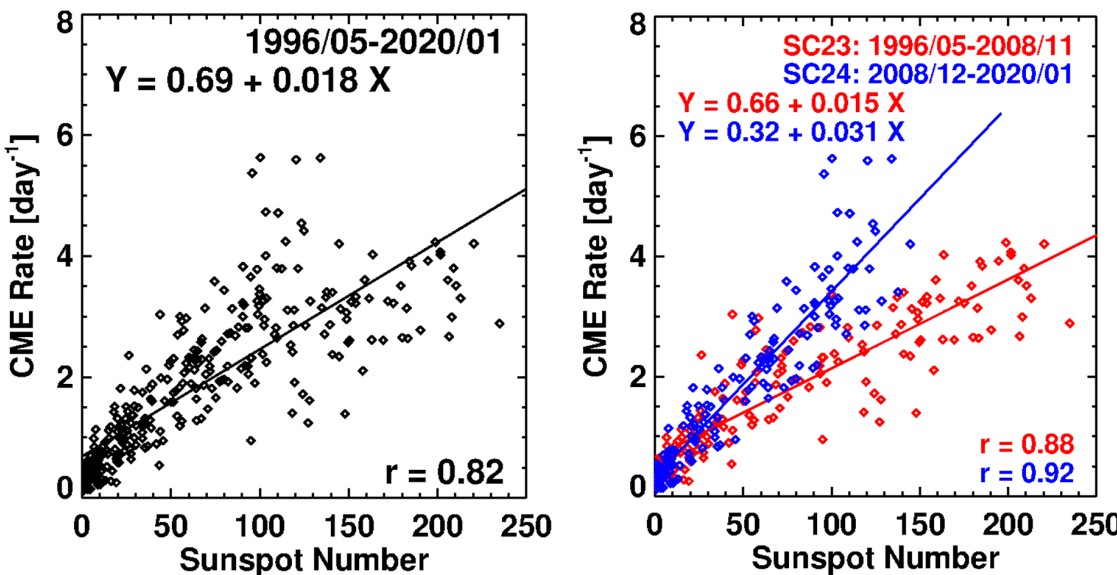

**Figure 20.** Scatter plot between the sunspot number and CME occurrence rate for the whole SOHO observing period until the beginning of the year 2020 (**left**) and separately for cycles 23 and 24 (**right**). The lower correlation on the left plot is because the relationship changed in cycle 24 compared to cycle 23. For example, the high rate around SSN = 100 is entirely due to cycle-24 CMEs. For a given SSN, the CME rate is much larger in SC 24. For SSN = 100, the regression lines indicate a CME rate of 3.42 in SC 24, compared to 2.16 in SC 23. The scatter plot will be revisited after checking whether CME identification made by different people might have affected the CME rate, especially those with widths close to 30°.

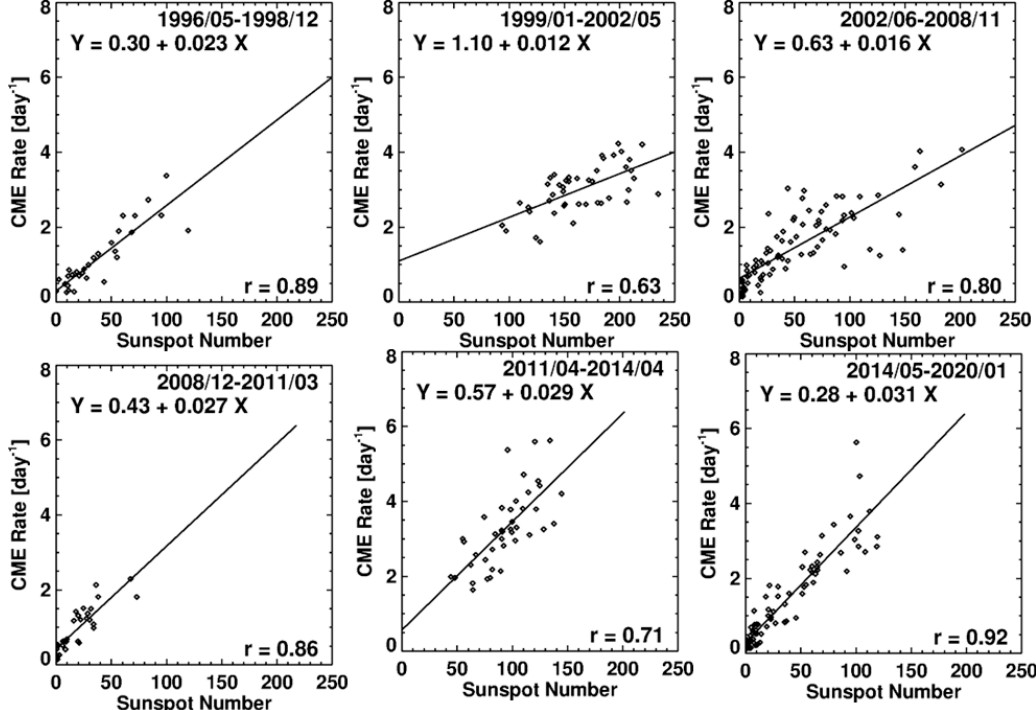

**Figure 21.** Scatter plot between the sunspot number and daily CME rate in solar cycles 23 (**top row**) and 24 (**bottom row**). The left, middle, and right columns give the scatter plots in the rise, maximum, and declining phases, respectively. The correlation coefficients and regression lines are shown on the plots. Note that the maximum phases in the two cycles have the lowest correlation between the SSN and CME rate.

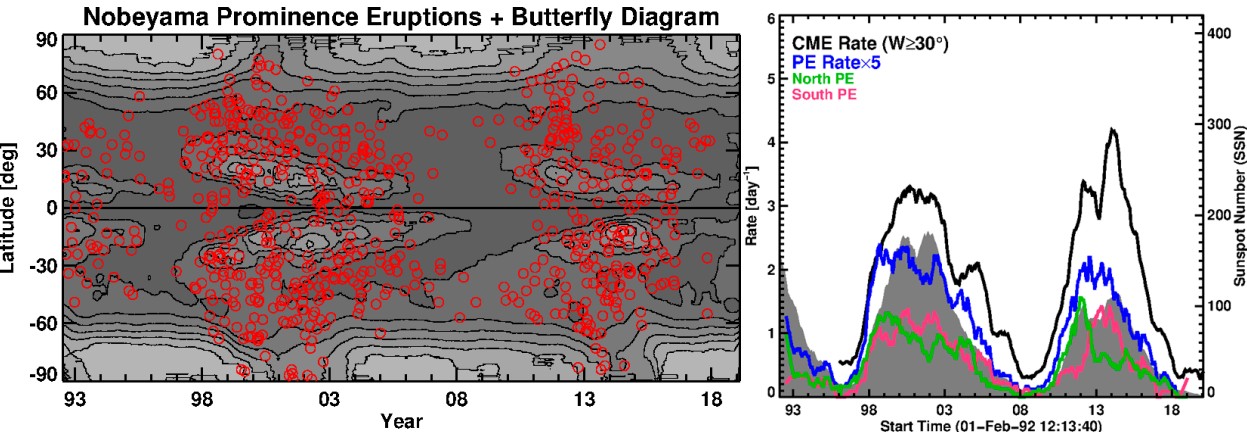

**Figure 22.** (**left**) Microwave butterfly diagram (contours) showing the 17 GHz brightness temperature at low latitudes due to active regions and at high latitudes due to the polar magnetic field. The red circles represent locations of prominence eruptions detected automatically in 17 GHz images of the Sun. (**right**) Comparison between CME and prominence eruption (PE) rates. The PEs from the Northern and Southern Hemispheres are distinguished by different colors. The gray plot in the background is the smoothed sunspot number. The CME and PE rates have been smoothed over 13 Carrington rotations.

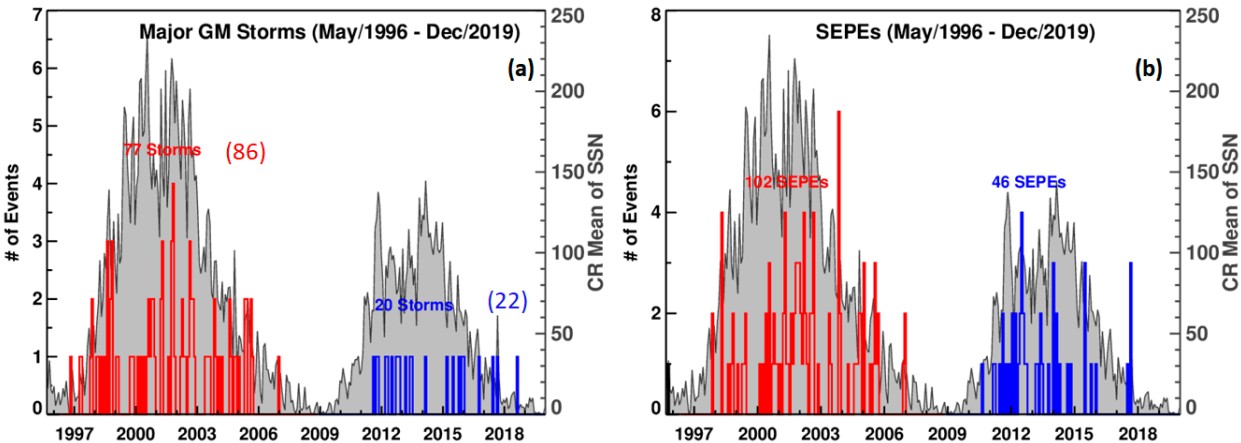

**Figure 23.** # means Number in Figure 23. The number of CME-associated major geomagnetic storms (**a**) and large SEP events (SEPEs) (**b**) summed over Carrington rotation periods. In (**a**), the numbers in parentheses are the total number of storms including those due to CIRs. SSN is shown in gray.

## 9. Extreme Space Weather Events

The mild space weather discussed in the previous section is a case of extreme event. In the opposite end of the spectrum, there are large events in the recent history as well as in the natural archives, such as tree rings and polar ice cores (see [78] for a review). A rough idea on the size of extreme events can be obtained by looking at the cumulative distribution of known events. Figure 24 shows the cumulative distributions of event sizes in large SEP events and intense geomagnetic storms. The distributions are obtained using modern data available in the space age. From the distributions, we can see that the one-in-100 year and one-in-1000 year SEP events have sizes of $\sim 2 \times 10^5$ pfu and $1 \times 10^6$ pfu, respectively. The largest event plotted in Figure 24a occurred on 23 March 1991 and has a size of $\sim 4.3 \times 10^4$ pfu [189], about five times smaller than a 100-year event. The 23 July 2012 event was estimated to have a similar size observed at STEREO ahead [157]. The tree-ring event of AD 774 identified by Miyake et al. [190] is indeed a 1000-year event. Inspired by this event, further investigations have resulted in several tree-ring and ice core events that qualify for a 1000-year event [78].

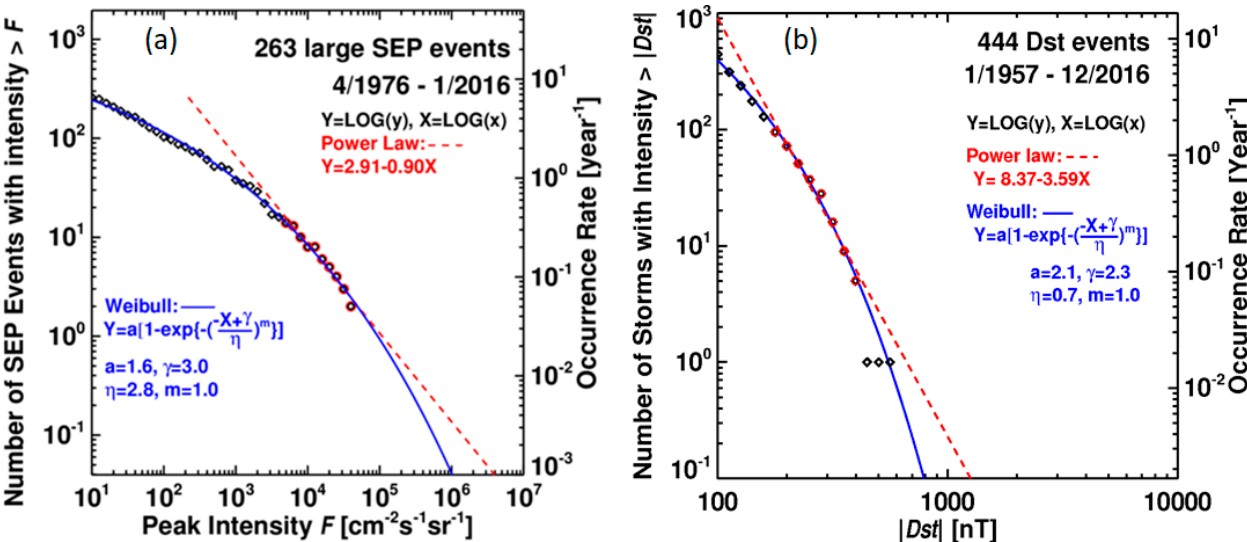

**Figure 24.** (**a**) Cumulative distribution of large SEP events from 1976 to 2016. (**b**) Cumulative distribution of intense geomagnetic storms (Dst ≤ −100 nT) and their yearly rates using Dst data from 1957 to 2016. In both cases, Weibull and power law fits to the distributions are shown. The right side Y-axis gives the yearly rate of the events. The curves are extrapolated to an occurrence rate of $10^{-3}$ per year, which gives the size of a one-in-1000 year event. The Weibull and power law distributions can be used to estimate event sizes that are more extreme. Adapted from [191].

The cumulative distribution of intense geomagnetic storms is based on the Dst index recorded since 1957. The Weibull fit to the cumulative distribution shows that 100-year and 1000-year event sizes are −603 nT and −845 nT, respectively. The 14 March 1989 storm is the largest event (Dst = −589 nT), plotted in Figure 24b, which is clearly a 100-year event. The intensity of the Carrington 1859 event has been estimated to be between −850 nT [192] and −1600 nT [193], indicating that it clearly is a 1000-year storm. Several new events have been identified based on sightings of low-latitude overhead auroras: about six 100-year storms and three 1000-year storms have been identified over the past five centuries [78]. Extreme SEP events and geomagnetic storms are exclusively due to energetic CMEs. Figure 25 shows the cumulative distribution of CME speeds with the average speeds of various CME populations. CMEs associated with purely metric type II bursts have a speed of only ~600 km/s, still faster than the general population (~400 km/s). On the other hand, GLE-associated CMEs are the fastest (~2000 km/s). Halo CMEs and CMEs associated with magnetic clouds (MC), non-cloud ejecta (EJ), and IP shocks (S) are similar to those associated with geomagnetic (GM) storms. This is understandable because all these CME populations indicate CMEs directly impacting Earth and causing GM storms. The next two populations are CMEs causing decameter-hectometric (DH) type II bursts and large SEP events. These events are due to electrons and ions accelerated by CME-driven shocks, similar to GLE events but accelerated to lower energies. This figure shows that a couple of thousand CMEs with speed exceeding ~600 km/s have significant space weather consequences. Figure 25a also shows that the number of CMEs drops rapidly for speeds >2000 km/s. The drop is modeled using Weibull and power law functions, as shown in Figure 25b. The Weibull distribution indicates that 100-year and 1000-year CMEs have a speed of 3800 km/s and 4700 km/s, respectively. The highest-speed data point in Figure 25b is from the 10 November 2004 CME that had a speed of 3387 km s$^{-1}$, close to a 100-year event. The corresponding kinetic energies are $4.4 \times 10^{33}$ and $9.8 \times 10^{33}$ erg, which are only a few times greater than the highest reported values [191]. These limits are ultimately decided by the limiting strengths of solar active regions that are determined by the maximum field strengths in the convection zone [194].

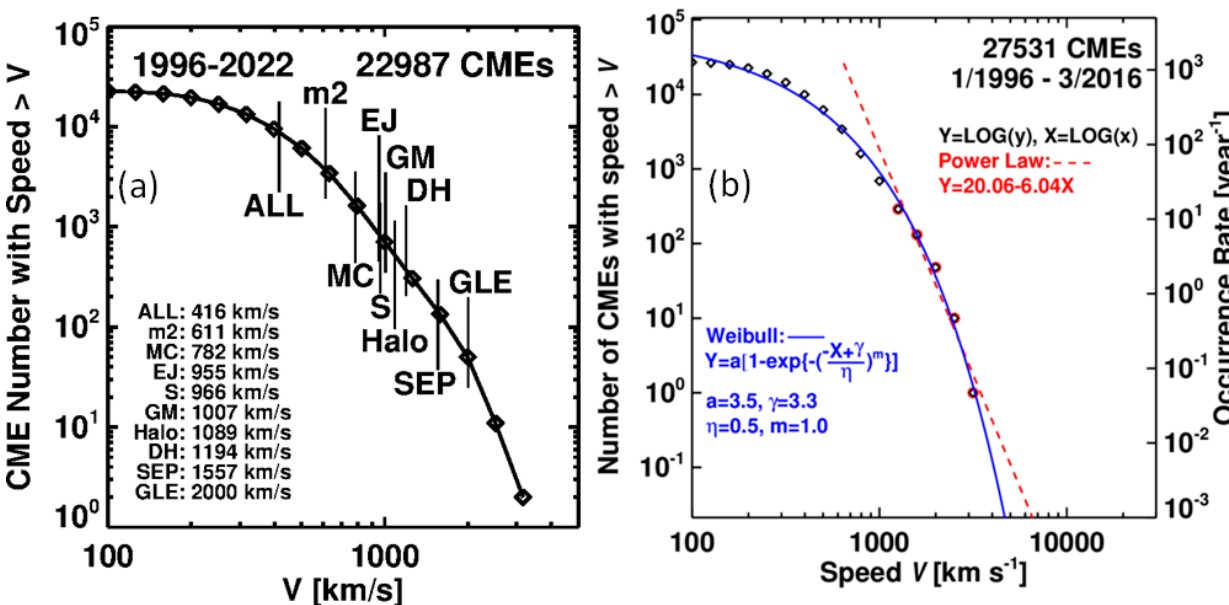

**Figure 25.** (**a**) Cumulative distribution of CME sky–plane speeds (V) from https://cdaw.gsfc.nasa.gov/CME_list, (accessed on 1 October 2022), with the average speeds of various CME populations. The original data is binned into 5 data points per decade. (**b**) Weibull (blue) and power law (red) fits to the CME speed distribution. Updated from [191].

## 10. Concluding Remarks

We have a fairly good understanding of large space weather events that can be linked to solar eruptions from magnetically closed regions and coronal holes. The electromagnetic component of solar eruptions is the solar flares that have prompt response in the form of magnetic crochet and sudden ionospheric disturbances. It takes only ~8 min for the X-ray photons to reach Earth's atmosphere. Closely following flares are GLEs that are delayed only by a couple of minutes. CME-driven shocks, on the other hand, take anywhere from half a day to a few days to reach Earth and cause SSC or sudden impulse. Thus, the timescales involved range from minutes to a few days. Accordingly the predictability of flares, SEPs, and CME/shock arrival differ significantly [195]. While predicting flare/CME occurrence based on source region properties is a long way away, there are methods being developed actively using statistics and machine learning (see [196] and references therein). Predicting SEP events also needs to be probabilistic in nature [197,198]. Methods of predicting SEP occurrence based on neural networks are being actively pursued (see e.g., [199] and references therein). The statistical result that spacecraft anomalies peak a couple of days after the start of an SEP event (see Figure 17) can be used to predict certain impacts after detecting a GLE event, because the GLE and SEP intensities are correlated [200]. Similarly, predicting the shock arrival (SSC) at 1 au [201] has value, because spacecraft anomalies also peak a couple of days after SSC (see Figure 18).

We have mainly considered the space weather consequences of solar eruptions and coronal holes. We have not considered CME initiation or the trigger of eruptions [202–205]. The discussions in this paper are concerned with the mature stage of CMEs after the completion of magnetic reconnection in the eruption region [206]. The mature flux rope that follows after the initial seed flux rope becomes unstable and initiates the flare reconnection [207]. The seed flux rope can be hot [208] or cold (associated with a filament) [209,210]. We have also not discussed another aspect of CMEs and CIRs, viz., the Forbush decrease (see [211–213] and references therein), nor the overall increase in cosmic ray flux due to weak solar activity [214].

The solar and heliospheric community has made enormous progress in understanding solar magnetic variability and its impact on the inner heliosphere, especially on Earth's space environment. The progress can be attributed to the rapid advances in space missions

that culminated in SOHO and STEREO. Recent observatories, such as the Parker Solar Probe and the Solar Orbiter, have started aiding deeper investigations of solar variability. Data from multiple views of the Sun from vantage points away from the Sun–Earth line (L4 and L5, polar orbit), coupled with global MHD modeling, are expected to rapidly advance our knowledge of solar magnetic variability.

**Funding:** This research was funded by NASA's Living With a Star program.

**Institutional Review Board Statement:** Not applicable.

**Informed Consent Statement:** Not applicable.

**Data Availability Statement:** All data used in the article are publicly available and the URLs provided.

**Acknowledgments:** This review article benefited greatly from NASA's open data policy in using SOHO, STEREO, and SDO data. I thank S. Yashiro and P. Mäkelä for help with some figures and S. Akiyama for formatting the references according to the journal style.

**Conflicts of Interest:** The author has no conflict of interest.

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
