# Peer review of "The Sun and Space Weather"

_atmosphere, doi:10.3390/atmos13111781_

Round 1
Reviewer 1 Report
The review paper 'The Sun and Space Weather’ summarizes the observational results on CMEs and HSS, and their relevance to space weather conditions, which effects can strongly disturb the Earth’s magnetosphere, ionosphere and atmosphere. These solar phenomena are the main causes of geomagnetic storms and acceleration of high energetic particles, which are dangerous to humans and to the technological structures in space, and can drastically disturb the magnetosphere and ionosphere conditions affecting the surface and spatial technologies based on radio communication.
The work is well organized and comprehensively described, presenting clearly the concepts and results, which are well referenced.
So the review is relevant in the context to improve the understanding in the connection between solar variability and space weather conditions, presenting what has been done and the actual state of art in this research area.
So in my opinion, the paper needs only minor revisions to be acceptable for publication.
Minor revisions
Line 40: Use ‘A solar flare represents’ instead of ‘A solar flares represents’
Line 122: Since it is first time MC appears gives the definition ‘magnetic cloud’
Lines 135-139: in the Figure 2 legend is this text ‘bins (-50 to +50)? [Seiji: Fix the scale. What is the number of data points in the acceleration plot? Any restrictions (4 data points minimum?].’ correct?
In Figure 4 the inside Med values might be like 3.5 x 10 +14, instead of 3.5e+14, I think.
Line 206: the title of section 3, I think is 'CME source regions, flares and filaments' instead of `CME source regions, Flares, and CMEs '
Line 361: Dst (Disturnbance storm time)
Line 408: change `an SIR` by `a SIR`
Line 451: `the axial filed is` is `the axial field is`
Line 491: please revise the text ` typical CMEs, viz, 38⁰).`
Lines 623-625: the figure caption must be revised. The explanation of (a) and (b) are not in agreement with Iucci et al., 2005 ((a) Normalized frequency of anomalies, averaged over the first 2 days of proton enhancements, at different orbits (red line, HH; green line, HL; and black line, LH) dependent on maximal flux of protons with energy E > 10 MeV (IMP 8). (b) Probability of anomalies for the high-altitude and high-inclination group of satellites dependent on the value of maximum flux of protons with E > 10 (black line) and E > 60 MeV (red line) (IMP 8)).
Line 637: `electron flux is enhanced` instead of `electron flux in enhanced`
Line 685: ` from open field regions` instead of ` from open filed regions`
Line 785: ` The 23 July 2012` instead of ` The 2012 July 2012`
Line 861: ` points away from` instead of ` points sway from`
Line 1015: the year of the Cerruti et al reference is 2006 instead of 2008
Author Response
I thank the referee for the positive comments and pointing out typographical errors. The errors are fixed as suggested.
Line 40: Use ‘A solar flare represents’ instead of ‘A solar flares represents’ OK
Line 122: Since it is first time MC appears gives the definition ‘magnetic cloud’ OK
Lines 135-139: in the Figure 2 legend is this text ‘bins (-50 to +50)? [Seiji: Fix the scale. What is the number of data points in the acceleration plot? Any restrictions (4 data points minimum?].’ correct?
Sorry, this material was supposed to be deleted. Now removed.
In Figure 4 the inside Med values might be like 3.5 x 10 +14, instead of 3.5e+14, I think.
Either notation is OK, but the exponential notation avoids superscripts so the figure does not look crowded.
Line 206: the title of section 3, I think is 'CME source regions, flares and filaments' instead of `CME source regions, Flares, and CMEs '
Good point. corrected.
Line 361: Dst (Disturnbance storm time) OK
Line 408: change `an SIR` by `a SIR` OK
Line 451: `the axial filed is` is `the axial field is` OK
Line 491: please revise the text ` typical CMEs, viz, 38⁰).` OK
Lines 623-625: the figure caption must be revised. The explanation of (a) and (b) are not in agreement with Iucci et al., 2005 ((a) Normalized frequency of anomalies, averaged over the first 2 days of proton enhancements, at different orbits (red line, HH; green line, HL; and black line, LH) dependent on maximal flux of protons with energy E > 10 MeV (IMP 8). (b) Probability of anomalies for the high-altitude and high-inclination group of satellites dependent on the value of maximum flux of protons with E > 10 (black line) and E > 60 MeV (red line) (IMP 8)).
The figure caption is revised as suggested.
Line 637: `electron flux is enhanced` instead of `electron flux in enhanced` OK
Line 685: ` from open field regions` instead of ` from open filed regions` OK
Line 785: ` The 23 July 2012` instead of ` The 2012 July 2012` OK
Line 861: ` points away from` instead of ` points sway from` OK
Line 1015: the year of the Cerruti et al reference is 2006 instead of 2008 OK
Reviewer 2 Report
This review article by Nat Gopalswamy, summarizes major milestones in understanding the connection between solar variability and space weather.
The authors give sufficient review and good methodology.
Only minor point needs to be modified:
In abstract and acknowledgments sections, the word “review” should be added before the word “article”
Author Response
I thank the reviewer for the positive comments.
I have taken care of the comment - In abstract and acknowledgments sections, the word “review” should be added before the word “article”